# The role of atmospheric large-scale patterns for recent warming periods in Greenland from 1900-2015

Florina Roana Schalamon[1], Sebastian Scher [1,2], Andreas Trügler[1,3], Lea Hartl[4,5], Wolfgang Schöner[1] and Jakob Abermann[1]

[1]Department of Geography and Regional Sciences, University of Graz, Heinrichstr. 36, 8010 Graz, Austria
[2]Wegener Center for Climate and Global Change (WEGC), University of Graz, Brandhofgasse 5, 8010 Graz, Austria
[3]Know-Center, Research Center for Data-Driven Business and Artificial Intelligence, Sandgasse 34/2, 8010 Graz, Austria
[4]Institute for Interdisciplinary Mountain Research, Austrian Academy of Sciences, Innrain 25, 6020 Innsbruck, Austria
[5]Alaska Climate Research Center, University of Alaska Fairbanks, 2156 Koyukuk Drive, Fairbanks, AK 99775, USA

*Correspondence to*: Florina Roana Schalamon (florina.schalamon@uni-graz.at)

**Abstract.** Atmospheric large-scale patterns strongly determine Greenland's regional climate through air mass advection and local weather conditions, making them essential to understand atmospheric variability. This study analyses the occurrence of atmospheric large-scale patterns during two distinct warming periods of the recent past that we identify objectively in climatological data. The first warming period lasted from 1922 to 1932 and an average air temperature increase of 2.9 °C across all stations considered for this study. The second warming period lasted from 1993 to 2007 and had an average warming of 3.1 °C. We apply Self-Organizing Maps as a clustering technique based on the geopotential height of the 500 hPa pressure level using 20CRv3 reanalysis data to characterize prevalent atmospheric large-scale patterns and investigate their occurrence, persistence, and effects on air temperature anomalies at our study site (Qaamarujup Sermia) in West Greenland. Both warming periods show similar overall air temperature anomalies. However, the distribution of large-scale atmospheric patterns differs significantly, while the relationship between atmospheric large-scale patterns and local air temperature seems to be constant in time. This suggests that variations in Greenland's warming are influenced by shifts in atmospheric circulation. This study emphasizes the critical role of changes in atmospheric large-scale patterns for understanding Greenland's warming periods.

## 1 Introduction

Warming periods (WPs) have played a critical role in shaping Greenland's climate and environmental systems. These periods of sustained temperature increase significantly influence the Greenland Ice Sheet (GrIS) and its contribution to global sea level rise. Box et al. (2009) described the history of air temperature (AT) over Greenland from 1840 to 2007 and identified one WP from 1919 to 1932 and another one from 1994 to 2007. Their study shows that these warming trends are not uniform across seasons, with winter temperatures exhibiting much greater variability than summer temperatures. Near-surface AT in Greenland significantly impacts the length and intensity of the melt season, which is crucial for the GrIS's mass balance (Zhang

et al., 2022). Changes in AT can influence Greenland's ice dynamics through feedback mechanisms linked to surface albedo. Rising temperatures lead to reduced snow and ice cover, increased exposure of bare ice and land, the formation of melt ponds, and progressive darkening of the snow and ice surface due to melting and the accumulation of impurities. These processes all lead to a lower albedo and cause additional heat absorption and ice melt. This self-reinforcing cycle amplifies regional warming

and contributes to sea level rise, further illustrating Greenland's key role in global climate change. GrIS already accounted for an estimated sea level rise of $10.8 \pm 0.9$ mm (Shepherd et al., 2019). Future melting of the GrIS could add additional 5 to 33 cm sea level equivalent until the year 2100, depending on climate scenarios (Aschwanden et al., 2019). This further highlights the importance of monitoring AT over Greenland, particularly during WPs.

Observational data from WPs prior to 1961 and with that the recent AT increase (1991 onwards) is rare (Hanna et al., 2012).

Observations with high temporal and spatial resolution from Alfred Wegener's last expedition to Greenland in 1930 and 1931 provide unique insights into historic temperature development in West Greenland (Abermann et al., 2023). The combination of the historic and modern datasets gives the opportunity to investigate centennial-scale climate variability and its drivers. Using the same location as Wegener's expedition allows for a direct comparison between past and present atmospheric conditions, enabling a better understanding of long-term changes and the role of large-scale atmospheric patterns (LSPs) in

shaping regional climate variability. LSPs influence local and regional weather conditions by determining the advection of air masses with different intrinsic characteristics. Variability in AT, moisture content, and vertical movement affect precipitation patterns and impact radiative processes (Loikith et al., 2019). Cloud radiative processes, modulated by cloud height, optical thickness, and hydrometeor phase, are additional key drivers of the local energy balance and interact with LSP-induced atmospheric variability (Wang et al., 2018). Extreme weather conditions are closely linked to the occurrence, persistence, and

maximum duration of LSPs (Horton et al., 2015). As the Arctic region is warming at a pace more than double, up to four times that of the world average, a process known as "Arctic Amplification" (Rantanen et al., 2022; Taylor et al., 2023), Greenland serves as an exceptional case study to explore the dynamic role of LSPs in influencing climate trends. Understanding how LSPs modulate regional and local climate variability is critical to comprehending their broader impacts on atmospheric systems.

Two widely used atmospheric indices are the Greenland Blocking Index (GBI) and the North Atlantic Oscillation (NAO). The GBI reflects variations in the atmospheric pressure patterns by gauging the geopotential height at 500 hPa over Greenland (Barrett et al., 2020; Hanna et al., 2016). These fluctuations influence the blocking or redirection of westerly flows across the North Atlantic, significantly affecting regional AT and weather systems. The NAO reflects a redistribution of atmospheric mass between the Azores High and the Icelandic Low, capturing shifts in the strength and position of these pressure systems.

It is typically represented by surface pressure differences but can also be identified using geopotential height anomalies, as its influence extends throughout the troposphere. The NAO describes climate variability in the North Atlantic sector, influencing temperature and precipitation patterns across Europe, North America, and North Africa (Hanna et al., 2022; Hurrell et al., 2003; Silva et al., 2022).

Hanna et al. (2022) have extended historical records of these indices, analysing their trends and variability dating back to 1800. Their findings reveal the correlation between these indices and the occurrence of extreme weather events in northwest Europe as well as their impact on the sensitivity and response of the GrIS to global warming. An observed rise in the frequency and intensity of Greenland blocking during summer months 1991-2020 has significant consequences for AT patterns and weather phenomena in the Arctic, influencing the likelihood of extreme weather conditions in this rapidly warming region as they found out. Horton et al. (2015) found a relation between changing patterns of geopotential height in the northern hemisphere and extreme temperature.

To identify LSPs it is common to look at the geopotential height of the 500 hPa pressure level. Air advection can be inferred from the geopotential height fields in a sense that large scale air advection follows lines of equal height, with low heights to the left in the northern hemisphere. While this approach is commonly used to interpret synoptic-scale flow, it is important to note that thermal advection—i.e., the transport of warm or cold air—more precisely follows thickness gradients rather than height lines. Thus, patterns in geopotential height can indicate both flow direction and relative atmospheric temperature. In Greenland, the main upper-level atmospheric flow is from the southwest in winter and from the west in summer (Cappelen and Drost Jensen, 2021). Other synoptic patterns involve the preferred track of cyclones northward through the Davis Strait and Baffin Bay or move northwards following the west coast.

Despite recent advances in understanding Greenland's LSPs, knowledge gaps remain regarding LSP characteristics over centennial timescales, particularly concerning how their occurrence and influence may differ between historical and recent WPs. This limits our understanding of whether current LSP trends reflect stable or changing impacts on Greenland's local AT anomalies.

The primary goal of this study is to investigate the role of LSPs in shaping Greenland's regional temperature variability by comparing two distinct WPs in the last century. We aim to address two central questions: (A) How do the distributions of LSPs differ between WPs, and (B) what role do these patterns play in influencing the local AT at a specific study site in West Greenland? To explore these questions, we apply a Self-Organizing Map (SOM) algorithm to find clusters in reanalysis data from 1900 to 2015 on a daily timescale, focusing on the relative occurrence, persistence, and AT impact of individual LSPs across the study period. By clarifying the influence of LSPs on Greenland's AT variability, this study aims at advancing our understanding of atmospheric drivers in Arctic climate dynamics over centennial scales.

## 2 Data

### 2.1 Weather Stations

The observations used in this study consist of historical records digitized by the Danish Meteorological Institute (DMI) (Cappelen et al., 2021). We analysed data from five coastal stations: Upernavik (UPV), Nuuk (NUK), Ilulissat (ILU), Qaqortoq (QAQ), and Tasiilaq (TAS), as shown in Fig. 1 (a). These stations were selected because they provide monthly records starting as early as 1873, 1807, 1784, 1807, and 1895 respectively. A continuous record at all these stations is available from 1895

onwards. All stations are situated in settlements along the coast and spread around Greenland. TAS is the only station on the east coast. More detailed information can be found in Cappelen et al. (2021).

The region of interest of the project is seen in Fig. 1 (b) and shows the location of the weather station WEG_L. It is at 940 m a.s.l. and on the outlet glacier Qaamarujup Sermia connected to the GrIS (Abermann et al., 2023). This location is referred to as study site in the following.

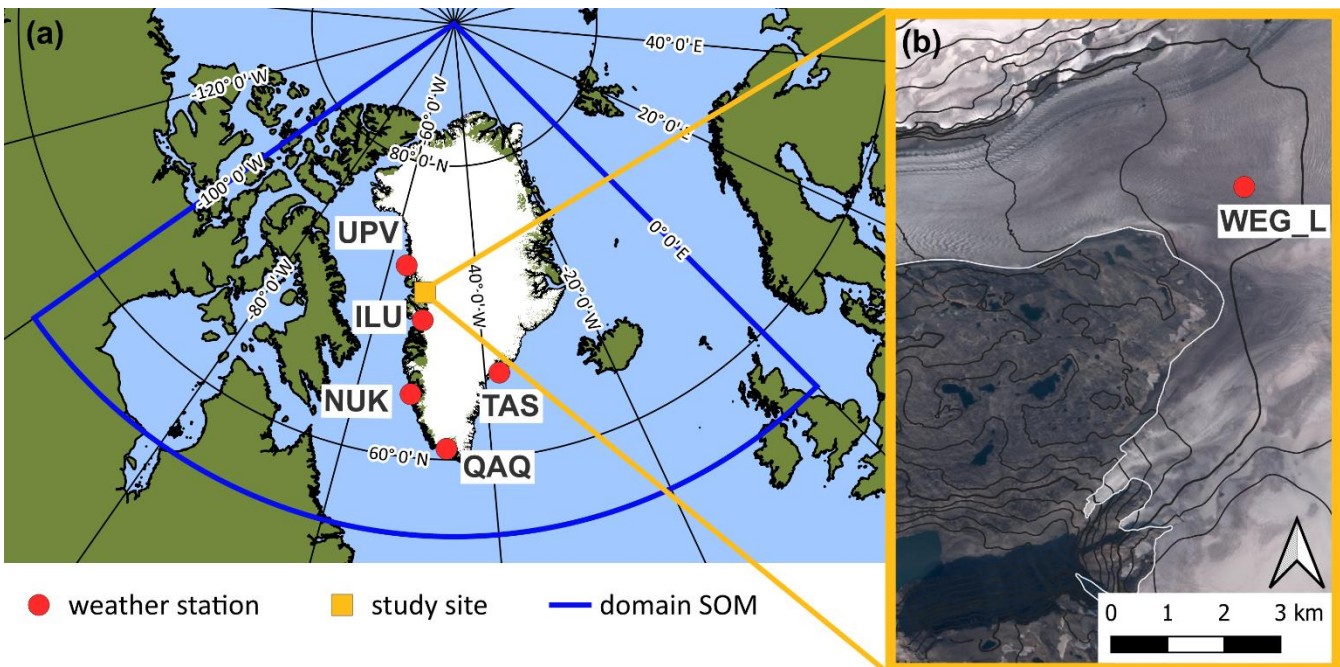

Figure 1: (a) Overview map of Greenland with the locations of the weather stations from DMI (red dots) and the study site (yellow square). The blue frame shows the domain used for the SOM analysis. (b) Detailed map of the study site Qaamarujup Sermia with the location of the weather station WEG_L. Created based on QGreenland v1.0.1. (Moon et al., 2021).

### 2.2 Reanalysis Data

We used reanalysis data from NOAA's 20CRv3 20[th] century reanalysis (20CRv3), provided by the NOAA Physical Sciences Laboratory. This historic reanalysis products only assimilate surface pressure, sea surface temperature and sea ice distribution to provide a best guess of atmospheric parameters. In contrast to other reanalysis systems that start from the satellite era (1970s onwards), 20CRv3 has a much longer temporal coverage. The dataset is available from 1836 to 2015 with an output every 3 hours and covers the whole globe at a resolution of 1°x 1°. It is available at 28 vertical pressure and 11 height levels (Compo et al., 2011; Slivinski et al., 2019).

20CRv3 was used for clustering the geopotential height of the 500 hPa pressure level to identify LSPs. It was also the source to compute spatial averages of AT for three different domains, namely for Greenland (6-75° W, 58-85° N), the Arctic (>66.5° N), and globally. For all areas the area weighted average AT including all grid points within the given domain was calculated.

The cosine of the latitude was used as weight to compensate for the smaller grid cells closer to the pole (Wei et al., 2022). We computed annual AT anomalies for each of the domains based on the mean of the period 1986-2015.

Additionally, 20CRv3 was used to generate the daily AT dataset at the study site. The approach follows the method described in Abermann et al. (2023). They evaluated the performance of the 20CRv3 and CERA-20C reanalysis models for two non-

assimilated stations within the study area. Both models were interpolated and adjusted to the station altitudes, and their findings indicate that 20CRv3 aligns more closely with observations from 1930 and 1931 than CERA-20C. For this reason, we selected 20CRv3 for our analysis. Slivinski et al. (2021) confirmed the general good agreement between 20CRv3 and measured temperature. The closest model grid points were linearly interpolated to the location and height corrected with 6.5 K/1000 m.

## 3. Methods

### 3.1 Warming Periods

In this study we focused on periods of increasing AT from the beginning of the 20[th] century up to the final year that 20CRv3 is available, thus the period 1900-2015. For this period, we investigated the AT anomaly at the weather stations with respect to the reference period, which we defined as the last 30 years of the study period (1986-2015). Our criterions for defining WPs

are 1) a continuous rise of 5 or more years of the AT anomaly in the 5-year running mean of annually AT anomalies and 2) an increase is time-synchronously apparent at all weather stations to exclude possible local drivers. To assess the magnitude of the warming a Sen's slope estimator was used (Sen, 1968). The estimator is the median of slopes of pairwise investigation of datapoints. Due to the use of medians, it is more robust to outliers than other methods. The significance of the trend was analysed with the Mann-Kendall test (Kendall, 1955; Mann, 1945).

Further, we compared the AT anomaly of the stations to three spatial averages-global, Greenland and Arctic-based on 20CRv3 and an extracted site-specific timeseries at WEG_L. For that the reanalysis product is linearly interpolated to the study site and height corrected following (Abermann et al., 2023).

Based on the 20CRv3 a spatial analysis was conducted to assess trends in WPs across the study domain. This analysis utilized the Sen's slope estimator to quantify the magnitude of trends and the Mann-Kendall significance test to evaluate their statistical

relevance. By applying these methods, we generated a spatial representation of the warming trends over the study area, providing insights into the variability and significance of AT anomalies across different regions.

### 3.2 Atmospheric Large-Scale Patterns

This section defines the methodology for identifying LSPs and analysing their impact on AT in Greenland. LSPs were identified using SOM algorithm, a well-established clustering method widely applied in climatological studies (e.g. Hartl et

al., 2020, 2023; Hofsteenge et al., 2024; Mattingly et al., 2018; Mioduszewski et al., 2016; Preece et al., 2022; Schmid et al., 2023; Schuenemann and Cassano, 2009). SOMs are a clustering method based on artificial neural networks (Kohonen, 2013)

and offer an automatic and reproducible framework for defining clusters in the geopotential height field. The identified LSPs were used to analyse their connection to local AT anomalies, particularly during WPs.

When clustering data with SOMs, a set of weight vectors is iteratively adjusted to represent input data points (Van Hulle,
2009). During training, the algorithm identifies the cluster centre with the weight vector closest to the input data, known as the "best-matching unit" (BMU). The BMU and its neighbouring cluster centres update their weight vectors based on the next input during the training phase. This causes the network to adapt and form clusters centres, creating a lower-dimensional representation of the input data. Just as with other clustering methods, cluster centres are identified based on a "training" dataset, resulting in a fitted SOM model. In the training dataset, every single sample is assigned to one of the cluster centres.
The fitted SOM model can then also be used on new data, and – without change in the definition of the cluster centres-assigns the new data samples to the cluster centres. The input data in this case was daily 20CRv3 geopotential height of the 500 hPa pressure level from 1900 to 2015. While alternative approaches using pressure anomalies could offer different insights into seasonality, we base our SOM analysis on absolute pressure fields to directly assess the influence of LSPs on regional warming. Each day in the data set was assigned to one of the cluster centres. In our study, the cluster centres represent LSPs, and we will
only refer to them as LSPs from now on.

To fit a SOM model, several parameters need to be selected: the number of cluster centres, the number of iterations during training, the learning rate and the distance function. We use the 1D SOM method by Doan et al. (2021), applying their distance function (structural similarity) and learning rate (0.1) as proposed by them. The structural similarity distance function has the advantage of being able to handle data with temporal and spatial structure (such as air pressure patterns in our case) better than
the commonly used Euclidian distance. To analyse 40 years of daily air pressure trends, Doan et al., (2021) performed 5000 iterations; however, because our study period is longer, we doubled the number of iterations to 10000.

The domain was defined to ensure comprehensive coverage of Greenland, placing the study site at the centre and accounting for key atmospheric influences. To capture the impact of prevailing westerlies and southwesterlies (Cappelen and Drost Jensen, 2021), the domain was extended westward, while the southern boundary was adjusted to include possible warm air intrusions
from the mid-latitudes. This resulted in a domain spanning 0-90° W and 55-90° N. Test runs with a larger domain (120°W - 20°E and 50-90°N) can be found in the appendix A1.

The selection of the number of cluster centres requires a balance between interpretability and representation, with the goal of achieving physically meaningful clustering of weather patterns. Choosing too few clusters risks oversimplifying the diversity of atmospheric phenomena, while too many hinder clear interpretations (Mioduszewski et al., 2016). Preece et al. (2022)
selected 12 clusters, Schuenemann and Cassano (2009) defined 35 clusters but later grouped them into six, and Schmidt et al. (2023) initially analysed 20 clusters before reducing them to four subgroups. We performed test runs for different number of clusters (see appendix A1), but we aimed to avoid the need for subgrouping after applying the SOM algorithm, opting instead for a straightforward and consistent clustering approach. We determined that eight clusters provided an optimal compromise and sufficient to represent distinct weather patterns without introducing overly rare LSPs.

To analyse the influence of LSPs on local AT and to compare the different WPs, we determined the LSP for each day between 1900 and 2015 and calculated the respective AT based on the linearly interpolated and height corrected 20CRv3 data at the study site WEG_L (see Fig. 1 (b)). Then the different characteristics of the LSPs were investigated: the relative occurrence, persistence, and the average AT anomaly. All are given for each LSP separately once for the whole study period 1900-2015, and then for each of the two WPs.

The relative occurrence of a LSP is the percentage of days with this certain LSP present, relative to the number of all days in the study period. We refer to the number of consecutive days with a certain LSP as persistence. The daily AT anomaly was computed with respect to the last 30 years of the study period (1986-2015) at the study site. It is based on the difference of the AT on that day compared to the average AT for the day of year with a centred running mean of 30 days. This is the AT anomaly with respect to a running climatology and are referred to as AT anomaly.

To test if there is a significant difference in the distribution of the relative occurrence of the LSPs between the study periods, a Chi-square test was performed. The Chi-Square test is appropriate for comparing observed frequencies of categorical data in a contingency table against expected frequencies under the null hypothesis of no association between the two distributions. That means in our case that the observed frequency of LSPs in one WP is compared to the expected frequency of the LSPs in the full study period. To ensure the result is robust, the SOM method was repeated 1500 times, starting again at the training

phase and then sorting each day into the defined LSPs. Each iteration was followed by testing for a significant difference in the distribution of relative occurrences.

We further investigated seasonal differences of the relative occurrence of the LSPs in the WPs relative to the occurrence in the entire study period 1900-2015. A value of 1 indicating an equal occurrence in the WP as in the full period while values above 1 represent a higher occurrence rate in the WP, and values below 1 indicate reduced occurrence.

An additional approach to investigating the relationship between LSP occurrence and local AT involves analysing the seasonal distribution of LSPs on the warmest and coldest days. To identify these extremes, all days from 1900 to 2015 were ranked based on AT anomaly values at the study site, selecting the top 15 % of days with the largest positive AT anomalies and the bottom 15 % with the largest negative AT anomalies.

## 4 Results

**4.1 Warming Periods**

The course of the AT anomaly between 1900 and 2015 relative to the reference period (1986-2015) at the stations UPV, ILU, NUK, QAQ and TAS, the 20CRv3 area average for the globe, the Arctic, Greenland as well as 20CRv3 interpolated to WEG_L shows two distinguished WPs (Fig. 2 (a)). These two periods are observed at all stations and show a continuous increase over more than 5 years. Based on this, we determine WP1 between 1922 and 1932, and WP2 between 1993 and 2007. During WP1,

the AT anomaly increased on average by 2.9°C across stations, while in WP2, it increased by 3.1°C, though WP2 spans a

longer period (14 years compared to 10 years for WP1). The average annual increase for both WPs across all stations is 0.2°C/year.

Other periods also show rising AT anomalies; however, these either last only 5 years or less (e.g., 1938-1943) or are not observed consistently across all stations. For example, from 1971 to 1980, the AT anomaly increases at TAS, while at other stations along the west coast of Greenland, the increase begins in 1975 and extends until 1981. Overall, UPV experiences the largest air temperature increase as the northernmost station.

A comparison of reanalysis data from WP1 and WP2 reveals distinct patterns of warming. During WP1, warming is concentrated over Greenland, whereas WP2 exhibits more globally widespread warming. This difference is evident when comparing the AT anomalies for Greenland, the Arctic, and global averages. In WP1, discrepancies between reanalysis data and observed temperatures are more pronounced, with area averages showing that Arctic AT anomalies even decrease at the beginning of WP1. By contrast, in WP2, all AT anomalies-globally, across the Arctic, and in Greenland-consistently increase, aligning closely with observations. Also, the extracted point timeseries at WEG_L follows the course of the AT anomaly of the observations.

Spatial analysis of the warming trends of two WPs, defined based on Greenland's AT anomaly of 20CRv3, further highlights these differences (Fig. 2 (b) and (c)). WP1 shows warming concentrated along Greenland's west coast and over the ocean between Greenland and Canada, with some regions even exhibiting a cooling trend. In contrast, WP2 demonstrates more uniform warming across Greenland and the surrounding regions, reflecting the broader extent of temperature increases during this period, The strongest warming is also seen over the ocean between Greenland and Canada, but much stronger than in WP1.

From a climatological perspective, WEG_L is located in a region where AT changes are among the strongest during both WPs, as seen in Fig. 2 (b) and (c). The particular strong and significant warming in this area further supports WEG_L as a representative study site, making it a well-suited choice for this study.

The seasonal analysis confirms the findings from Box et al. (2009)that the AT increase is strongest in winter (see Fig. A2.1)., while the smaller anomalies are observed in summer and autumn.

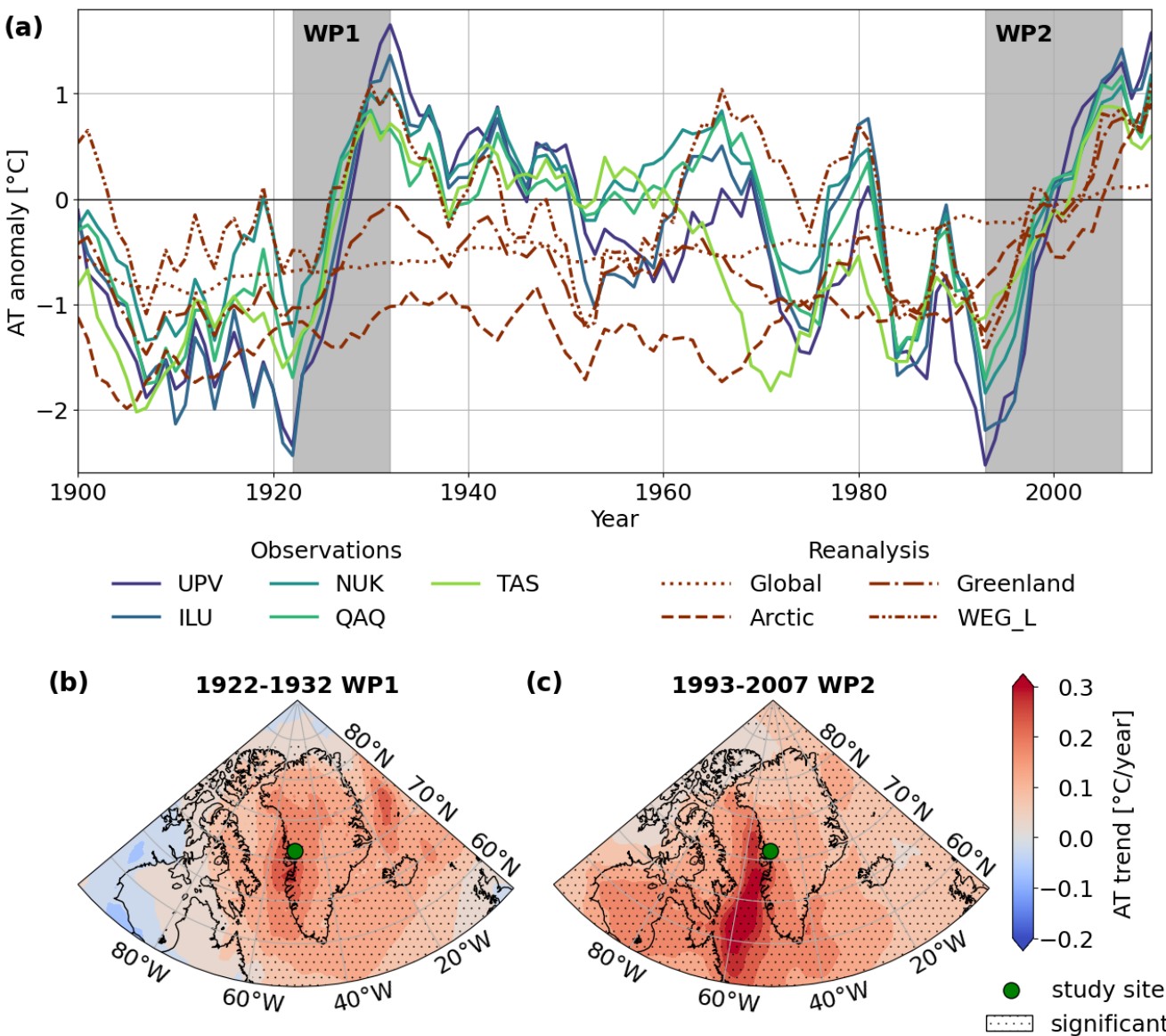

**Figure 2: (a): Annual AT anomaly with respect to reference period 1986-2015 at weather stations (Upernavik (UPV), Illulisaat (ILU), Nuuk (NUK), Qaqortoq (QAQ), Tasiilaq (TAS)), of 20CRv3 as spatial average of the Arctic, Greenland, globally and interpolated to the study site WEG_L, smoothed with a 5-year window rolling mean. The two defined WPs are marked with the grey background. (b) and (c): spatial representation of the Sen's slope estimator for the WPs. The colour shows the AT trend for one grid cell of 20CRv3, the "." hashing indicates grid cells where the trend is significant (Mann-Kendall test).**

### 4.2 Atmospheric Large-Scale Patterns

The LSPs obtained by clustering the large-scale reanalysis fields into eight clusters using SOM are shown in Fig. 3. Each of the eight LSPs shows distinctive individual features. Specifically, air advection towards the study site varies across the individual LSPs, with flow coming from the northwest, west, south, southwest and southeast. Additionally, we distinguish

different atmospheric conditions that influence the study site. There are cyclonic and anticyclonic patterns, based on the average geopotential height of the 500 hPa pressure level over the whole domain of 5297 m.

 LSP4, 5 and 8 are cyclonic patterns and LSP1 and 2 are anticyclonic patterns. LSP3 indicates a north-south gradient of the geopotential height of 500 hPa leading to zonal air flow over the study site. For both LSP6 and 7, the study site is between a
high- and low-pressure system leading to air being advected from the south to the study site. For LSP1 the air is coming from the northwest to the study site, for LSP2 and 3 it is from the west. The other LSPs advect air also from the south, except LSP5 when the study site is within the limits of the low-pressure area, indicating air is coming from the GrIS. Note that the numbering of LSPs (e.g., LSP 1, LSP 2, …) is arbitrary, as it is based on the first cluster identified during the SOM training process, which depends on randomly chosen initial input data.

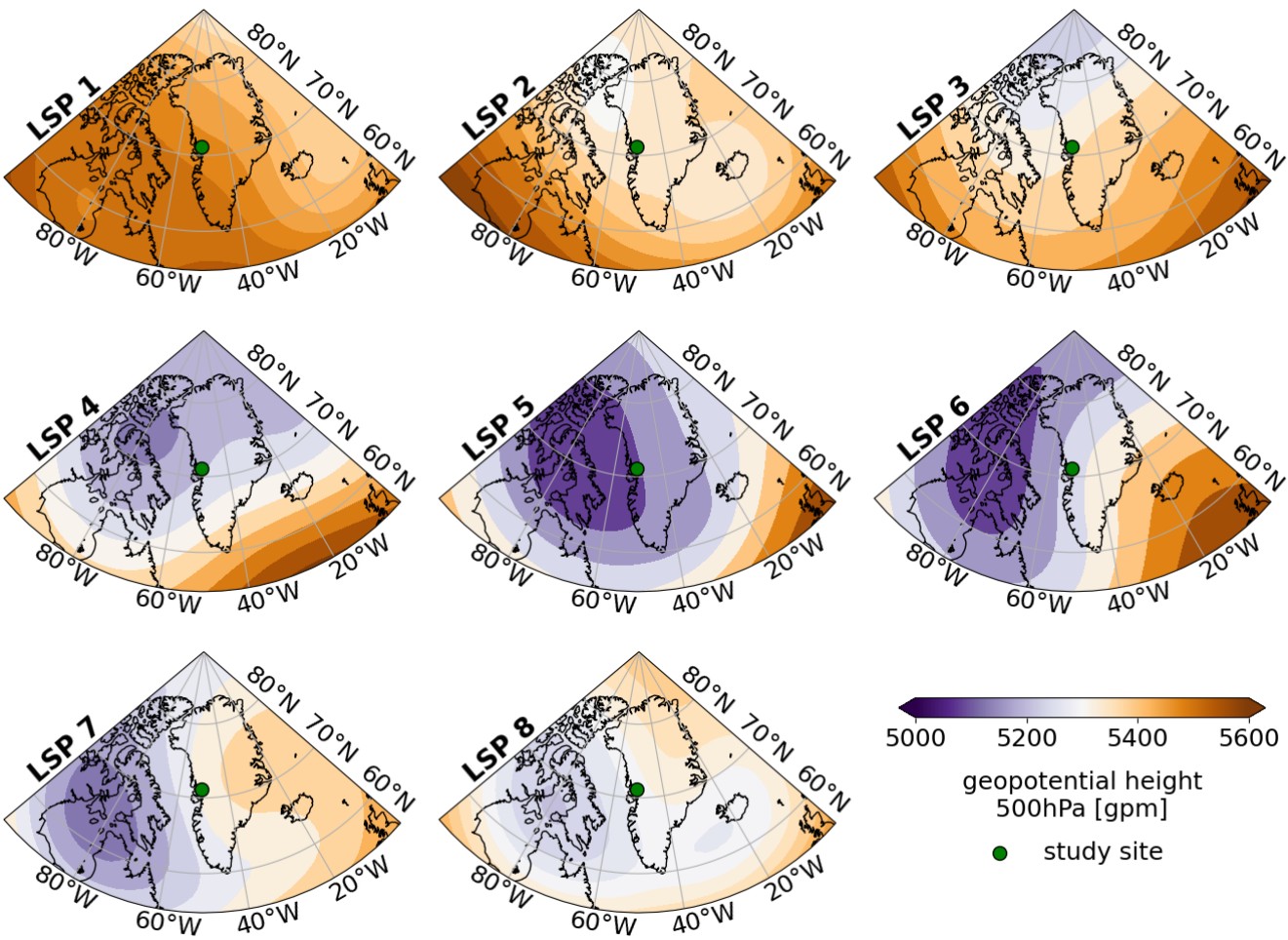


**Figure 3: Geopotential height of the 500 hPa pressure level of the eight LSPs as defined by SOM. The study site is marked with a green dot. For visual clarity, the SOM patterns are displayed in a 2D matrix; however, the underlying topology is one-dimensional, with neighbourhood relations applying only sequentially along a single line from the top left to the bottom right, i.e., following the numbering of the LSPs.**

In Fig. 4 (a) we see that LSP3 – and thus a zonal flow – is by far the most frequent pattern (47 % for the entire study period 1900-2015, 43.9 % for WP1, 47.2 % for WP2). LSP6 is the second most common pattern, with a consistent frequency across both WP1 (17.0 %), WP2 (17.1 %), and the whole study period (17.2 %). This steady occurrence of LSP6, along with a slightly higher frequency of LSP8 during the WPs compared to the full period, is a consistent feature of the two WPs compared to the whole study period. For the remaining LSPs, the occurrence in the WPs shows distinct variations from the long-term mean

occurrence. For instance, while LSP3 aligns closely with the frequency observed during the full study period and WP2, it is less prevalent in WP1. In contrast, LSPs 2 and 7 appear more frequently in WP1 than in either WP2 or the full period.

Differences between the WPs and the entire study period are small in general, between 0.1 % and 3.1 %. However, statistical testing (using a Chi-Square Test) reveals a significant difference in the distribution of LSPs among WP1, WP2 and the full study period. This result is also supported by various significance tests, including Fisher's Exact Test and the G-Test, as well

as alternative approaches to the standard Chi-Square test, such as the Monte Carlo and permutation-based resampling methods. The previously introduced robustness test with 1500 repetitions validates this result, making the significant difference robust. We did not find clear evidence of trends in LSP distribution per year. Figure A3.1 in the appendix provides further details, illustrating the annual relative occurrence of each LSP compared to its average occurrence over the full period (1900–2015) (a), as well as the relative occurrence per year (b).

To connect the LSPs to their potential local influence we analysed the average AT anomaly per LSP in the three study periods. Figure 4 (b) shows the mean as well as the standard deviation of AT anomalies per LSP. LSP6 and 7 have a positive AT anomaly in all three periods, but LSP6 has the largest standard deviation of all LSPs with instances when a negative AT anomaly is reached. LSP1, 3 and 8 contain the smallest total anomalies. LSP2, 4 and 5 are patterns with negative average AT anomaly. While LSP2 and LSP5 have on average air advection from either the cold GrIS or a northerly wind component, this

is surprising for LSP4 with advection from southwest (Fig. 3). Additionally, LSP2 and 4 have the largest differences between the WPs, so that average anomalies of LSP2 are warmer during WP1 whereas they are warmer in LSP4 during WP2. Overall, the average AT anomaly is similar between all three study periods.

Figure 4 (c) shows the LSPs' persistence (i.e., the length of consecutive days with the same LSP). The persistence distribution is plotted, as well as a vertical line for the average persistence. LSP3 has the longest average persistence of approximately six

days, corresponding to its high relative occurrence. LSP6 follows with the second-longest persistence of 4.1 days in WP1 versus 3.5 days in WP2 and 3.7 days in the full period. The longer persistence during WP1 is true for all patterns. Further investigations into whether AT anomalies increase or decrease with longer persistence (not shown) did not yield conclusive results.

To go into more details beyond the average of the study periods, Fig. 4 (d) presents the annual average AT anomalies per LSP

for the full period (1900–2015). LSP6 and 7 maintain positive AT anomalies throughout the entire study period, while LSP2 and 5 consistently show negative anomalies. LSP4 also has predominantly negative anomalies but includes years with positive anomalies, which are not concentrated in both WPs. The annual AT anomaly is colder at the beginning of the study period compared to the later years. LSP3 leads to small AT anomalies, averaging around 0°C, while LSP1 shows slightly greater

year-to-year variability. LSP8 stands out for its marked interannual variability, alternating between particularly warm years
(e.g., 6.4 °C in 1927; 7.4 °C in 1945) to particularly colder years (e.g., -10.6 °C in 1909; -8.0 °C in 1936).

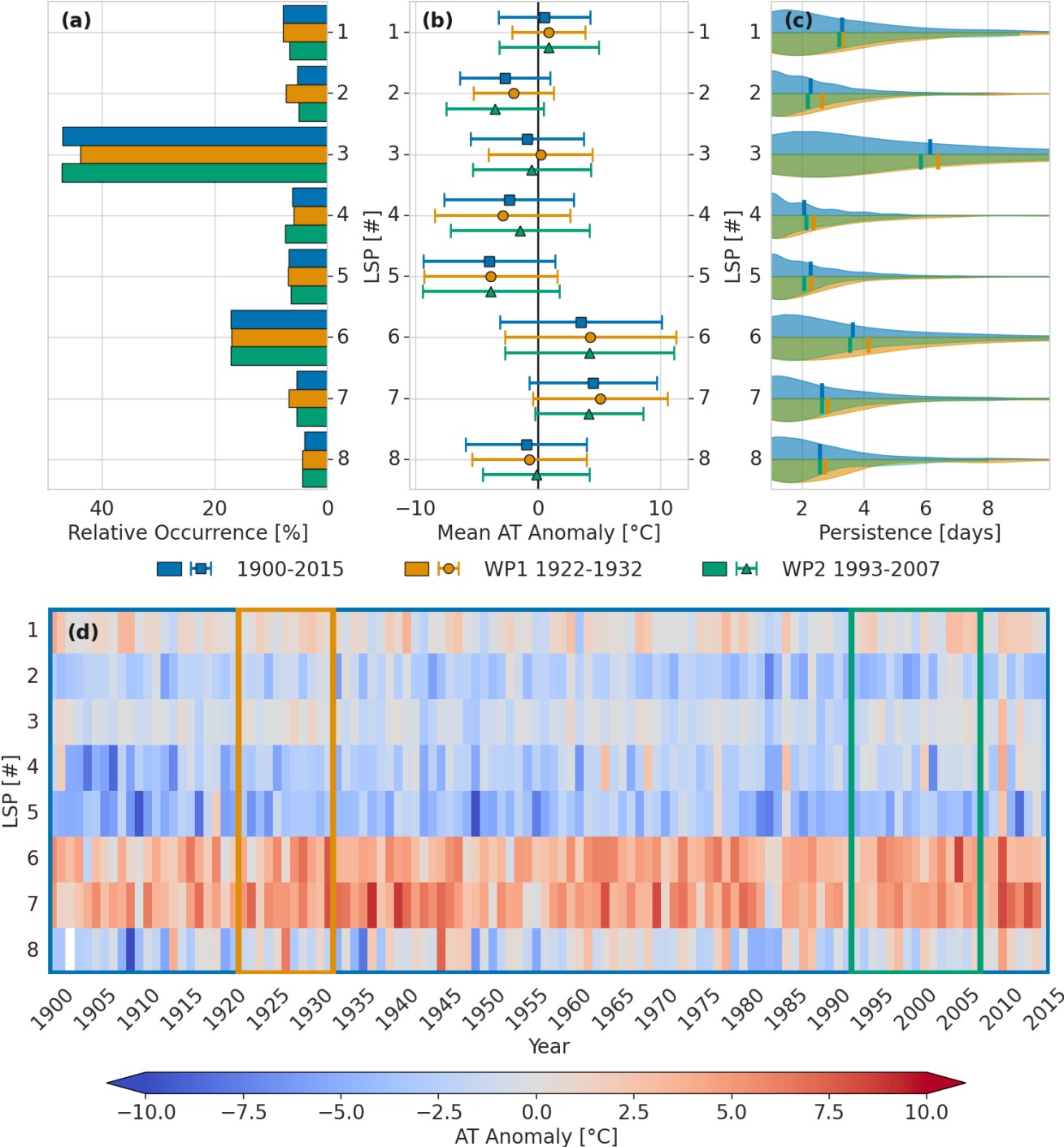

**Figure 4: Summary of the evaluation of the LSPs examined across the study periods. The study periods are color-coded throughout the plot as follows: WP1 (1922–1932) in orange, WP2 (1993–2007) in green, and the full period (1900–2015) in blue. (a) Distribution of relative occurrence of each LSP across the study periods. (b) Average AT anomaly per LSP, with markers indicating the mean**

anomaly and whiskers representing ±1 standard deviation. (c) Distribution of persistence in days per LSP, with bold lines indicating mean lengths. The full period is at the top and both WPs at the bottom. (d) Annual average AT anomaly per LSP, with coloured frames representing the study periods.


In addition to the year-to-year analysis, we extended our investigation to examine seasonal variations.

In a first step the relative occurrences were analysed seasonally and the full overview is displayed in Fig. A3.2. To highlight the small but significant differences between the occurrence of the LSPs, Fig.5 shows the occurrence in the WPs relative to the occurrence in the entire study period 1900-2015, split up by season. A value of 1 indicates an equal occurrence in the WP

as in the full period while values above 1 represent a higher occurrence rate in the WP, and values below 1 indicate reduced occurrence. This comparison reveals that certain LSPs have marked seasonal differences in their frequencies between WP1 and WP2. During WP1, LSPs 2 and 7 occur more frequently across all seasons. More days with LSP7, connected to a positive AT anomaly could lead to a warmer period, but LSP2, connected to negative AT anomalies could balance that. In WP2, LSP4 shows a higher frequency during all seasons, especially in spring and autumn, suggesting a shift in circulation dynamics

compared to WP1. LSP8 in WP2 shows an increased occurrence in spring, a season with notable AT anomalies in this period.

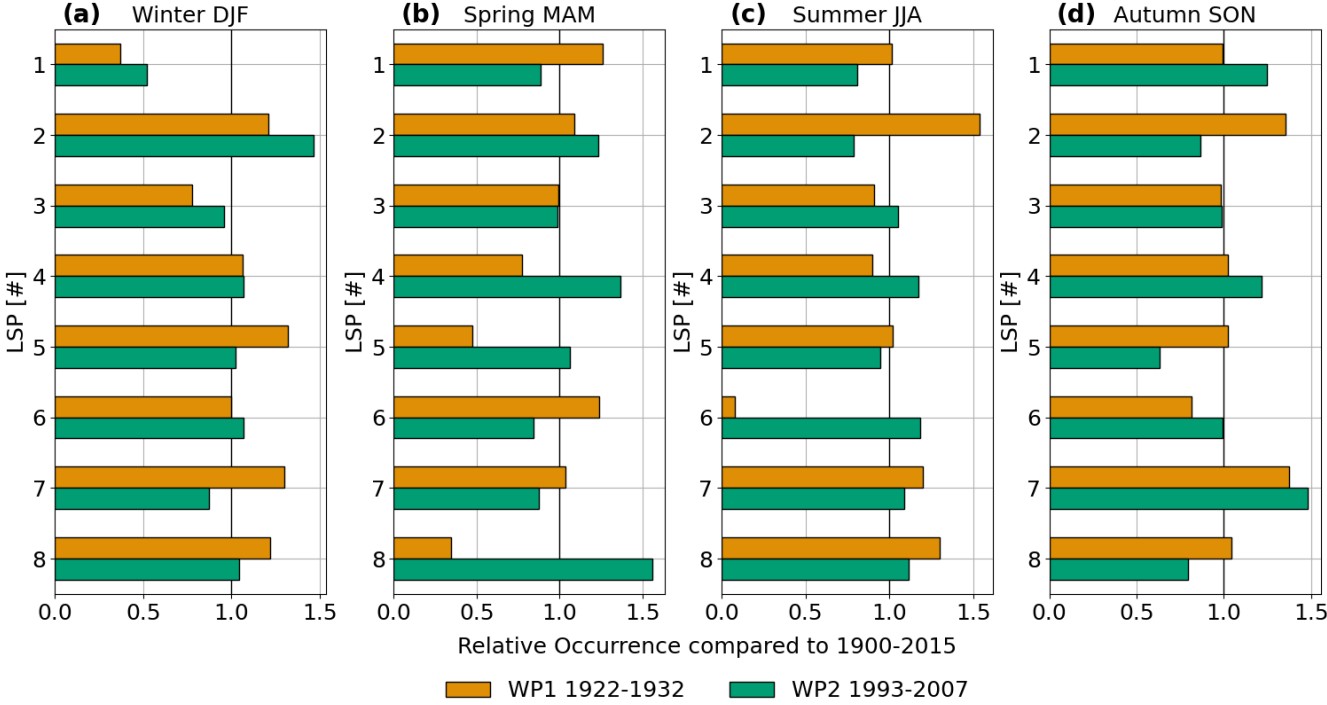

Figure 5: Seasonal relative occurrence of the LSPs in the WPs compared to the occurrence in the entire study period 1900-2015. A relative occurrence of 1 means the same relative occurrence, <1 less often, >1 more often. (a)-(d) show winter, spring, summer, and
autumn.

An additional approach to investigating the relationship between LSP occurrence and local AT involves analysing the seasonal distribution of LSPs on the warmest and coldest days. The relative distribution of LSPs within these two subsets was then computed (Fig. 6). We refer as the subset with the warmest days as warm days and the coldest days as cold days. The results for the two WPs and the full study period were similar, therefore, only the results for the full study period are shown here. The results for the WPs are shown in Appendix A4.

The analysis reveals distinct seasonal patterns in LSP occurrences during extreme AT anomalies. LSP3, the most common pattern, is predominantly associated with the coldest days across all seasons. However, for the warmest days, the seasonal associations vary. In winter, spring, and autumn, LSP6 is frequently linked to extreme warmth, while in summer, LSP3 is the dominant pattern. LSP1 contributes to some warm days but primarily during summer, when it is also connected to some cold days. LSP7 is exclusively associated with warm days, whereas LSP2, as well as LSPs 4 and 5, are primarily connected to cold days, with occasional instances leading to warm anomalies. LSP8 shows seasonal variability: in winter, spring, and autumn, it is linked to extremely cold days, but in summer, it is more frequently associated with warm anomalies.

The findings agree with the results shown in Fig. 4. LSP7 is primarily associated with positive anomalies, but it can also accommodate some negative days. LSP3 tends to dominate during cold days, though it is also linked to a majority of the warmest days, aligning with the annual temperature anomaly around 0°C (see Fig. 4 (c)). The seasonality of LSP8 could be connected to the variation of the annual temperature anomaly as shown in Fig. 4 (d). Additionally, LSP2, 4, and 5 correspond well with the annual temperature anomaly presented above.

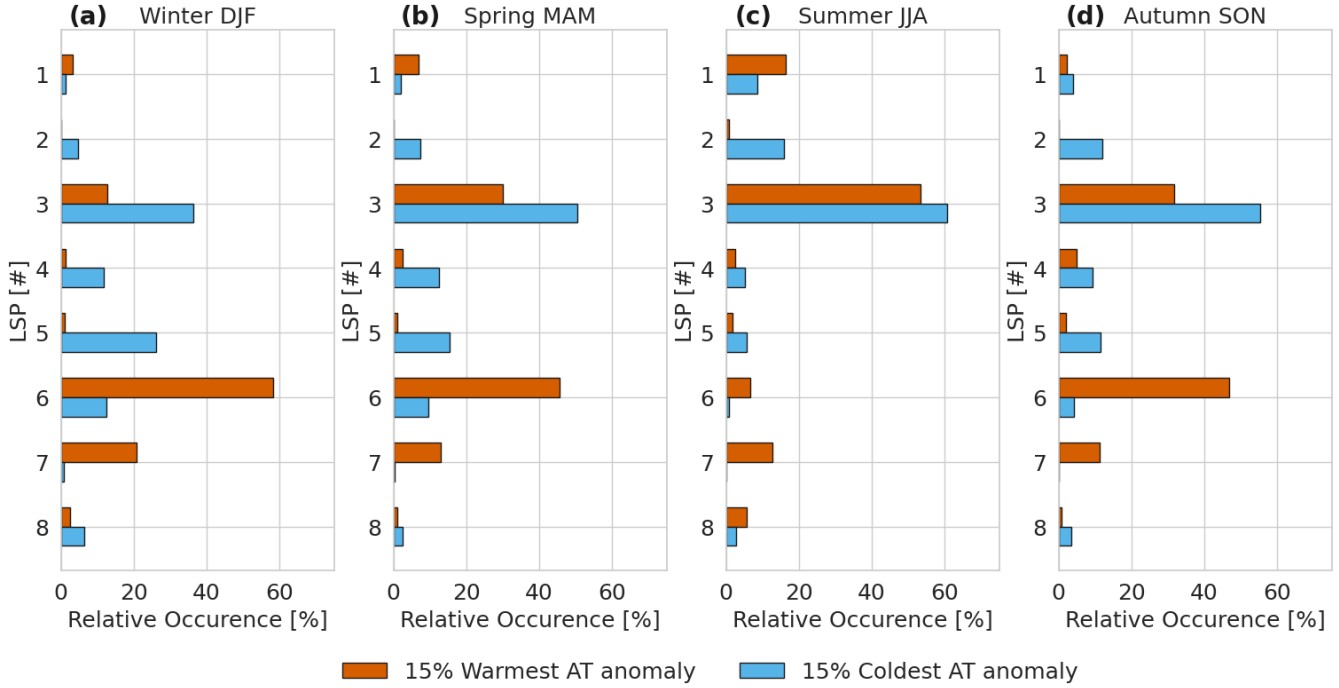

**Figure 6: Seasonal distribution of LSPs on the 15 % warmest and coldest days of the full study period 1900-2015.**

## 5 Discussion

### 5.1 Warming Periods

This study identified two WPs in Greenland between 1900 and 2015. WP1, occurring in the early 20[th] century, and WP2, spanning the late 20[th] to early 21[st] century. These findings align with established literature, such as the WPs identified by Box et al. (2009) and Łupikasza and Niedźwiedź (2019). Box et al. (2009) noted WPs from 1919 to 1932 and from 1994 to 2007, with more pronounced warming in South Greenland. While the exact timing of their WPs differs slightly from those in our study - likely due to differences in data sources, including their use of GrIS measurements compared to our focus on coastal stations - the general patterns remain consistent. Similarly, Łupikasza and Niedźwiedź (2019) identified early 20[th] century and late 20[th] century WPs in Svalbard that correspond broadly to WP1 and WP2. The slight differences in the exact timing and duration of their defined WPs are also because of their different research area.

When comparing area averages of 20CRv3 to a single station, differences is to be expected. The course of the extracted point timeseries at WEG_L shows that the usage of 20CRv3 aligns well with the course of the observations at the weather stations. A reason for the differences between reanalysis data and observed temperatures during the WPs could lie in sea ice extent parameterizations used in 20CRv3. Although 20CRv3 only assimilates surface pressure, it prescribes sea surface temperature and sea ice extent based on the HadISST2.3 dataset (Slivinski et al., 2019). The dataset applies different methods for estimating sea ice extent, with one approach used from 1900 to 1971 and another one from 1972 onward, along with an overlap of two sub-versions from 1981 to 2010. These differences in parameterization likely contribute to a more accurate representation of sea ice in WP2 compared to WP1, potentially resulting in a closer match between the area average of Greenland and observed AT anomalies in WP2.

### 5.2 Atmospheric Large-Scale Patterns

Choosing an appropriate number of cluster centres is key for successfully using SOM methods. Previous studies have taken varied approaches to this issue: Preece et al. (2022) selected 12 clusters, Schuenemann and Cassano (2009) defined 35 clusters but later grouped them into six, and Schmidt et al. (2023) initially analysed 20 clusters before reducing them to four subgroups. Our aim was general interpretability of the clusters in context with synoptic patterns during the WP and we chose a relatively low number of clusters form the start to avoid multiple clustering steps. The identified LSPs effectively capture the range of circulation scenarios expected over Greenland. These include westerly flows, as well as cyclonic and anticyclonic patterns, which are consistent with the known variability of atmospheric circulation in the region (Cappelen and Drost Jensen, 2021). The classification into eight distinct LSPs provides a robust framework for analysing their influence on local AT anomalies. The results highlight the significant role of LSPs in influencing local AT anomalies over Greenland and their variability across the different study periods. While the frequency distribution of LSPs shows relatively small differences between WP1, WP2, and the full study period, the statistically significant variations point to changes in atmospheric circulation associated with WPs.

The findings underscore that certain LSPs are more prevalent in specific WPs, such as the higher occurrence of LSP2 and 7 in WP1 and LSP4 in WP2. These shifts may reflect broader atmospheric changes during these periods, potentially linked to alterations in heat and moisture transport pathways. For instance, the increased occurrence of LSP4 during WP2, especially in spring and autumn, suggests evolving circulation dynamics that favour south-westerly advection, a mechanism that could indirectly amplify warming trends. The variability of LSP occurrences may also reflect broader atmospheric circulation changes, potentially influenced by climate modes such as the Arctic Oscillation (AO). A strong polar vortex, often associated with a positive AO phase, can contribute to more stable, zonal airflow patterns, while a weaker vortex during a negative AO phase allows for increased meandering and variability in geopotential height fields. These large-scale influences can provide additional context for understanding the observed shifts in LSP frequency and their role in shaping local AT anomalies.

The persistence of LSPs, especially during WP1, provides further insights. The longer average persistence of all LSPs during WP1 could amplify their impact on AT anomalies by sustaining certain atmospheric conditions. However, the lack of a clear relationship between persistence and the magnitude of AT anomalies suggests that the duration of a pattern alone does not dictate its influence. Instead, other factors, such as the interaction of LSPs with surface conditions (e.g., sea ice extent, GrIS surface characteristics) or broader climate changes may play a crucial role.

The consistent connection of certain LSPs with specific AT anomalies supports the stability of the relationship between LSPs and local climate conditions over time. For instance, LSP6 and 7 consistently yield positive AT anomalies, while LSP2 and 5 are tied to negative anomalies. The geopotential height field also reflects the mean virtual temperature of the tropospheric layer, where lower heights typically indicate colder air masses. For instance, the cold anomaly associated with LSP5 aligns with its position under a pronounced trough. Moreover, certain LSPs, such as LSP6, may represent configurations that support enhanced storm activity along common cyclone tracks—e.g., up Baffin Bay—which could contribute to temperature anomalies and variability along Greenland's west coast. The surprising negative AT anomaly of LSP4, despite its southwest advection, suggests the need for further analysis to understand the interplay between large-scale circulation and regional temperature responses, possibly incorporating additional variables such as cloud cover or precipitation patterns.

The variability observed in patterns like LSP4 and 8 can be understood with the influence of sea ice coverage, water vapor and large-scale circulation. For instance, during WP2, the lowered sea ice extent increased the water content of the atmosphere and modified the patterns of heat transport. This could intensify the warming effect typically associated with LSP4, especially during transitional seasons like spring and autumn. This could explain the lower annual AT anomaly for LSP4 at the beginning of the study period, when the sea ice extend was larger than today (Connolly et al., 2017). On the other hand, the high interannual variability of AT anomalies observed for LSP may reflect the influence of fluctuating sea ice conditions on atmospheric moisture availability, which subsequently impacts cloud cover and radiative fluxes. These dependencies demonstrate the necessity to complement LSP analysis with surface and atmospheric parameters, which are important for assessing climatic impact.

The seasonal analysis adds an important dimension to understanding LSP influences. The dominance of LSP3 during coldest days and warmest days in summer and its neutral AT anomaly overall indicate its role as a baseline pattern, with other drivers

dominating to influence the local AT. The marked presence of LSPs 6 and 7 during warmest days further confirms their role in driving positive AT anomalies, particularly during winter, spring, and autumn, showing that these patterns have a clear connection between the presence of the LSP and the effect on the local AT anomaly. Further the difference in warming between LSP6 (dominantly in winter) and LSP3 (dominantly in summer) can likely be attributed to the source of advection. During winter, LSP6 is associated with advection from warmer oceanic regions, while LSP3 in summer is influenced by advection from continental sources, which explains the seasonal variation in warming.

These findings align with earlier studies linking Greenland's AT to geopotential height patterns (Chen et al., 2016). They extend previous analyses by showing that the connection between LSPs and AT anomalies remains largely consistent across long timescales and different WPs. The persistence of large-scale atmospheric circulation patterns plays a central role in shaping storm tracks that influence accumulation and ablation over the GrIS. Persistent cyclonic or anticyclonic configurations can deflect or intensify the North Atlantic storm track, modulating both the frequency and intensity of moisture transport. These patterns impact surface mass balance not only by controlling snowfall during accumulation events, but also through warm-air advection, changes in cloud cover, and altered radiative forcing. Future studies should focus on integrating these findings with projections of LSP trends under various warming scenarios, incorporating climate indices, and examining surface-atmosphere interactions to refine our understanding of Greenland's temperature dynamics and their implications for Arctic climate feedbacks. While our selected domain effectively captures the dominant LSPs over Greenland, future studies could benefit from an expanded and adjusted spatial domain to better resolve specific influences-such as atmospheric rivers-that may significantly affect surface climate and mass balance.

Although the objectives of this study could be achieved, there are limitations to the analysis. One key limitation lies in the reanalysis data used; while 20CRv3 provides the most reliable long-term dataset available for this type of study, uncertainties inherent in historical reanalysis data may influence the results. Another limitation concerns the intrinsic variability of the self-organizing map (SOM) method, as the clustering outcomes depend on the choice of parameters, which are, to some extent, subjective. Although we followed established methodologies to mitigate biases, further refinement and comprehensive testing of SOM parameterization could enhance robustness. Finally, this study does not fully incorporate external factors, such as the AO phases or changing sea ice conditions, which are critical modulators of atmospheric patterns and temperature anomalies. While we discuss their potential interactions, a more detailed integration of these elements into future models is necessary to fully understand Greenland's warming dynamics and their broader climatic implications.

**6 Conclusions**

The first step of the study to define periods of increasing AT in Greenland results in two distinct WPs: WP1 (1922-1932) and WP2 (1993-2007), which generally correspond to prior research. Although there is some discrepancy between observations and 20CRv3, it is shown that the reanalysis data can be used for the following analysis, to investigate the influence of atmospheric LSPs on AT anomalies in Greenland across the WPs.

We identified eight different LSPs by applying a SOM algorithm on the geopotential height of the 500 hPa pressure level from the 20CRv3 historic reanalysis between 1900 and 2015. By analysing daily LSP occurrences, we identified significant differences in LSP distribution between the two WPs. While both WPs show comparable AT increases, they differ slightly but significantly in LSP occurrence, with WP1 having an increase of cyclonic patterns (e.g., LSP2 and 7) and WP2 of LSP4, a pattern with air advection from the southwest, suggesting shifts in circulation that may influence Greenland's climate response over time.

Despite these differences, the link between LSPs and temperature variability appears consistent across both periods, indicating that the mechanisms driving local temperature changes have remained relatively stable over time. The analysis revealed that while some LSPs are more common during the WPs, the frequency and persistence of these patterns alone do not fully account for the warming trends. Instead, our findings imply that the local AT is influenced by both the specific LSP characteristics and potentially other climate mechanisms, such as Arctic feedback loops, other factors detached from atmospheric drivers (e.g., sea ice occurrence), increase in $CO_2$ concentration and changes in global circulation. Notably, the LSPs associated with positive AT anomalies (LSP6 and 7) drive warm winter extremes, but the warmest days in other seasons are frequently linked to the most common pattern, LSP3, with a generally weak AT anomaly. This suggests that background warming may enhance an increase of AT regardless of LSP type, especially in WP2, which occurred during a period of more globally uniform warming, in contrast to WP1, where warming was more regionally concentrated in the Arctic.

It was not possible to quantify the net effect a change in LSPs distribution can have on the AT during WPs. These results underline the complexity of Greenland's climate response to warming and highlight the need for further research on the interplay between local, regional and global climate drivers. Understanding how LSPs interact with broader atmospheric changes, including shifts in sea ice extent and feedback processes, will be essential to predict Greenland's future climate accurately. Future studies could expand on this work by analysing seasonal effects in more detail, investigating the connection to climate indices (e.g., NAO and GBI), and projecting future LSP trends under different warming scenarios. A further application of our research will be to connect LSPs to measured and modelled surface mass balance of the glacier in the study area, as it is known that extreme melt events are closely linked to atmospheric (Fettweis et al., 2013; Hermann et al., 2020; Neff et al., 2014). Together, these approaches could provide a more comprehensive view of the mechanisms driving Greenland's warming and its implications for Arctic and global climate dynamics.

**Appendices**:

## A1. Consideration for the chosen SOM Parameters

In the following we show different parameters we tested for the SOM method. We tested different number of nodes – 6,7,8,9,10,15,20,25,30. As an example we added here the results with 20 nodes. As we wanted to avoid the manual regrouping

470 after SOM as done by Schmidt et al. (2023) and Schuenemann and Cassano (2009) we concluded that eight clusters are sufficient to show the expected large-scale patterns over Greenland and that they show the expected warm and cold patterns. More clusters do not result in more details. We further found that the key results are not sensitive to the number of clusters.

We tested different domain sizes for the SOM analysis, including domains extending further south to 30°N and even covering the entire Northern Hemisphere. However, these broader domains produced less meaningful patterns in the context of 475 Greenland, often introducing LSPs with little relevance to regional conditions. The domain selected for our main analysis— spanning 0–90°W and 55–90°N—captures the synoptic-scale circulation most relevant for Greenland. To illustrate the effect of domain choice, we also show results from a slightly extended domain (120°W to 20°E and 50°N to 90°N) using both eight and 20 SOM nodes. While using more clusters may offer finer distinctions, it also introduces interpretation challenges: for example, LSP 19 and 20 would need to be regrouped to allow for meaningful conclusions. Additionally, some patterns occur 480 very rarely—LSP 19 appears on only 138/12/26 days (0.3/0.3/0.5%) during the full study period (1900–2015), WP1, and WP2 respectively—limiting the statistical robustness of their associated AT anomalies.

That is why we opted for eight cluster centers and the selected domain, as this combination offers a robust and interpretable set of large-scale patterns that are both relevant for Greenland and statistically meaningful across the study period.

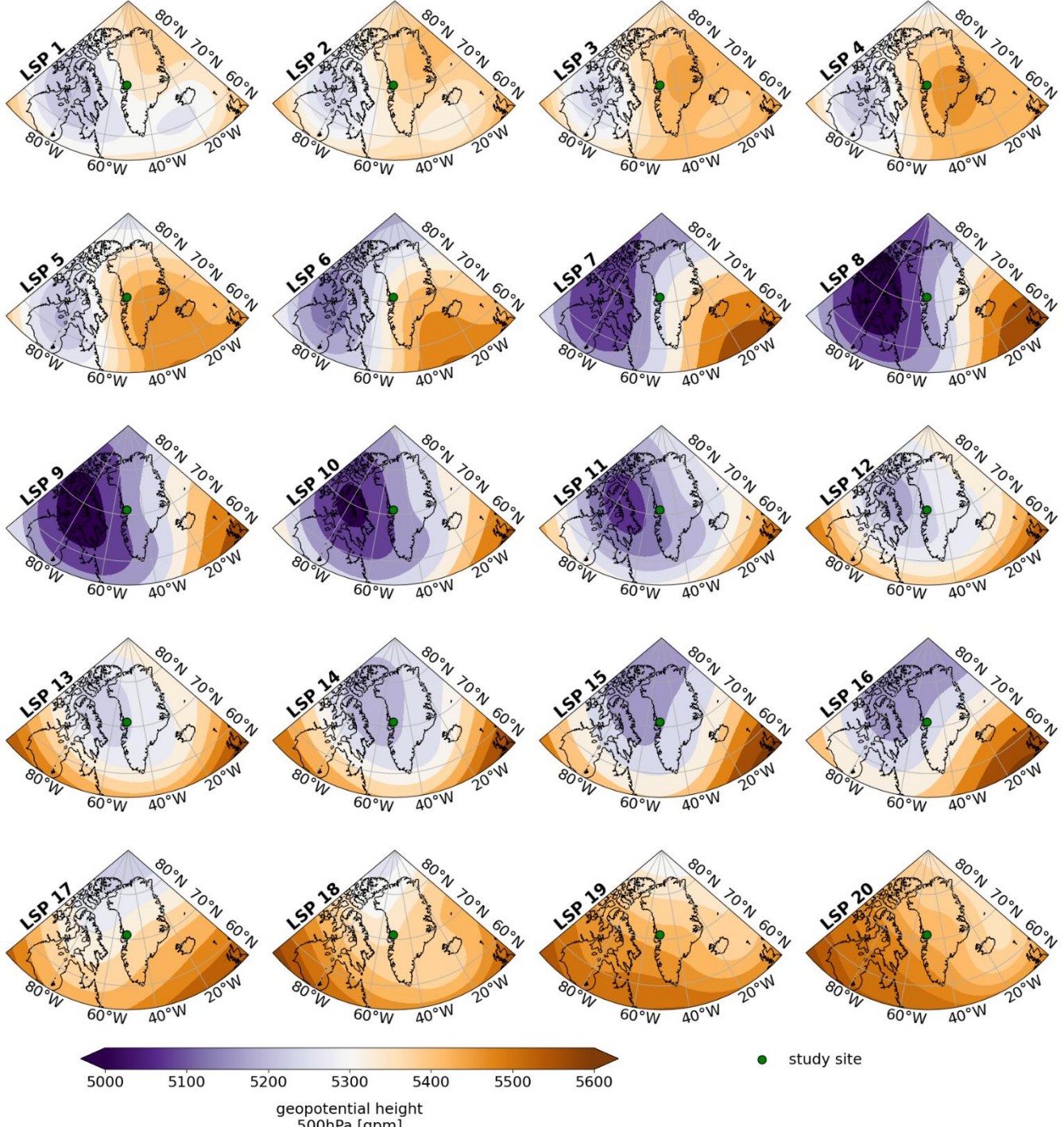

485

**Figure A1.1: Geopotential height of the 500 hPa pressure level of the 20 LSPs as defined by SOM with an input domain of (0-90° W and 55-90° N). The study site is marked with a green dot. For visual clarity, the SOM patterns are displayed in a 2D matrix; however, the underlying topology is one-dimensional, with neighbourhood relations applying only sequentially along a single line from the top left to the bottom right, i.e., following the numbering of the LSPs.**

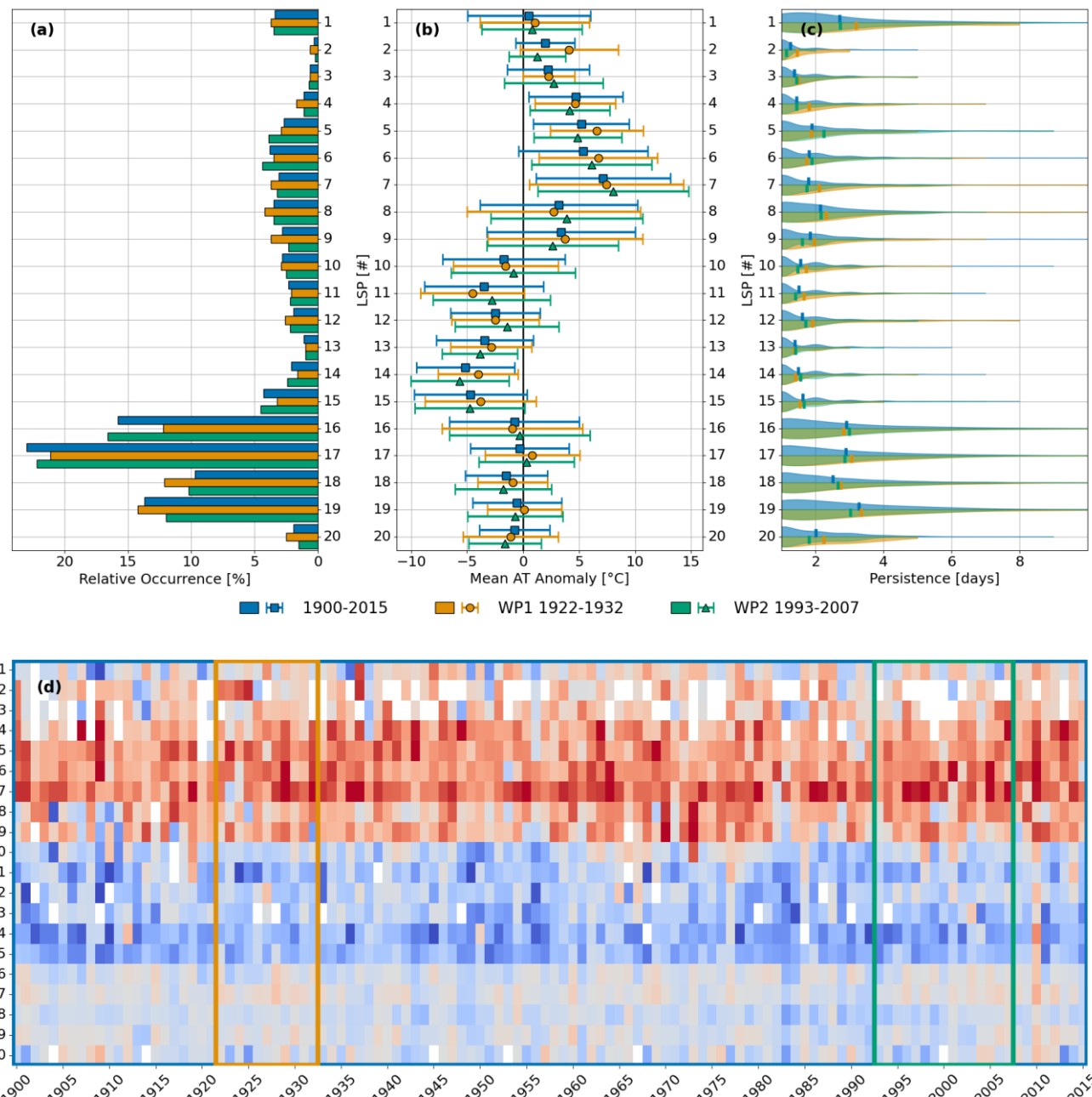

**Figure A1.2: Summary of the evaluation of the 20 LSPs examined across the study periods. The study periods are color-coded throughout the plot as follows: WP1 (1922–1932) in orange, WP2 (1993–2007) in green, and the full period (1900–2015) in blue. (a) Distribution of relative occurrence of each LSP across the study periods. (b) Average AT anomaly per LSP, with markers indicating the mean anomaly and whiskers representing ±1 standard deviation. (c) Distribution of persistence in days per LSP, with bold lines indicating mean lengths. The full period is at the top and both WPs at the bottom. (d) Annual average AT anomaly per LSP, with coloured frames representing the study periods.**

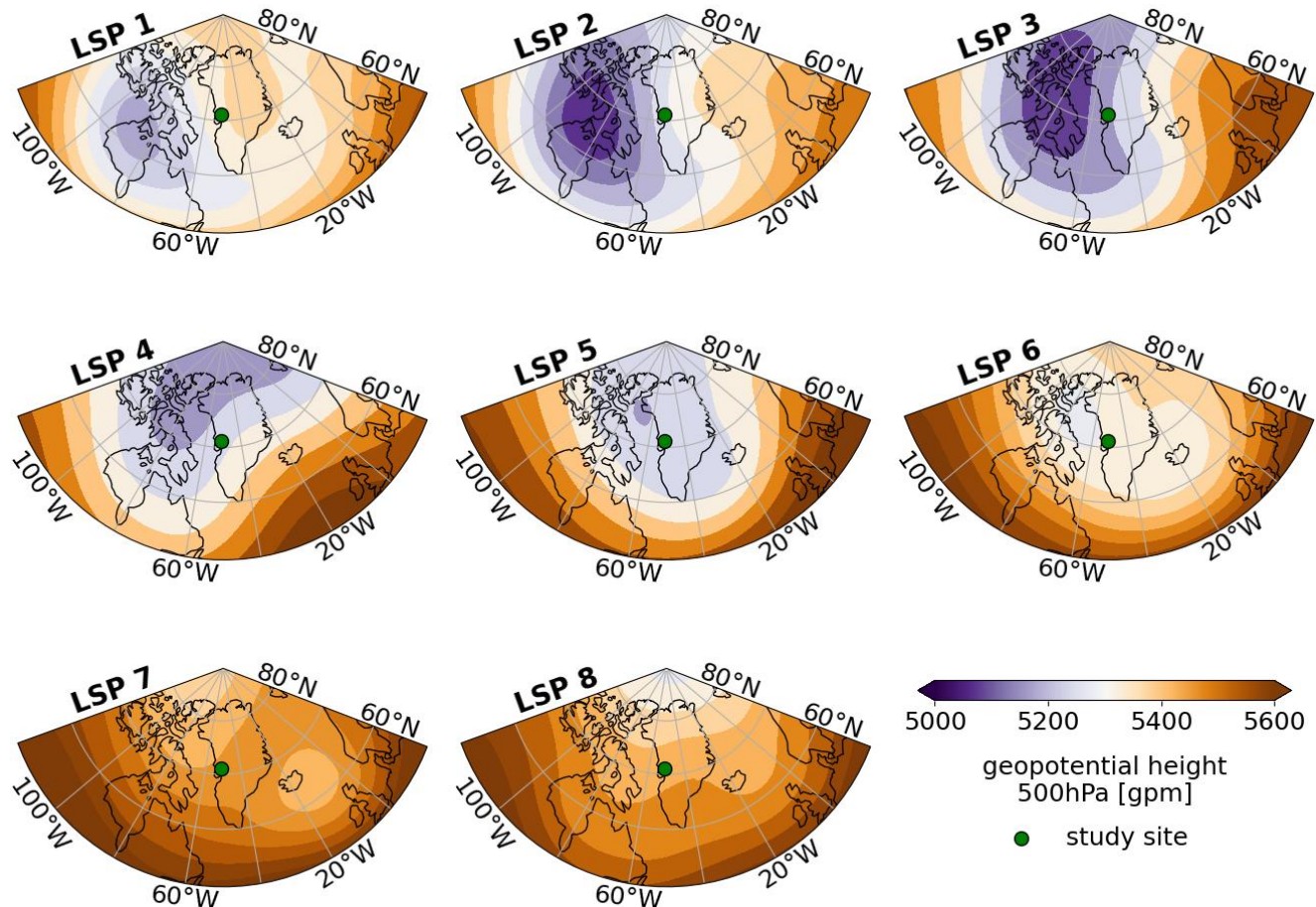

**Figure A1.3: Geopotential height of the 500 hPa pressure level of the eight LSPs as defined by SOM with a larger input domain of 120°W -20°E and 50-90°N. The study site is marked with a green dot. For visual clarity, the SOM patterns are displayed in a 2D matrix; however, the underlying topology is one-dimensional, with neighbourhood relations applying only sequentially along a single line from the top left to the bottom right, i.e., following the numbering of the LSPs.**

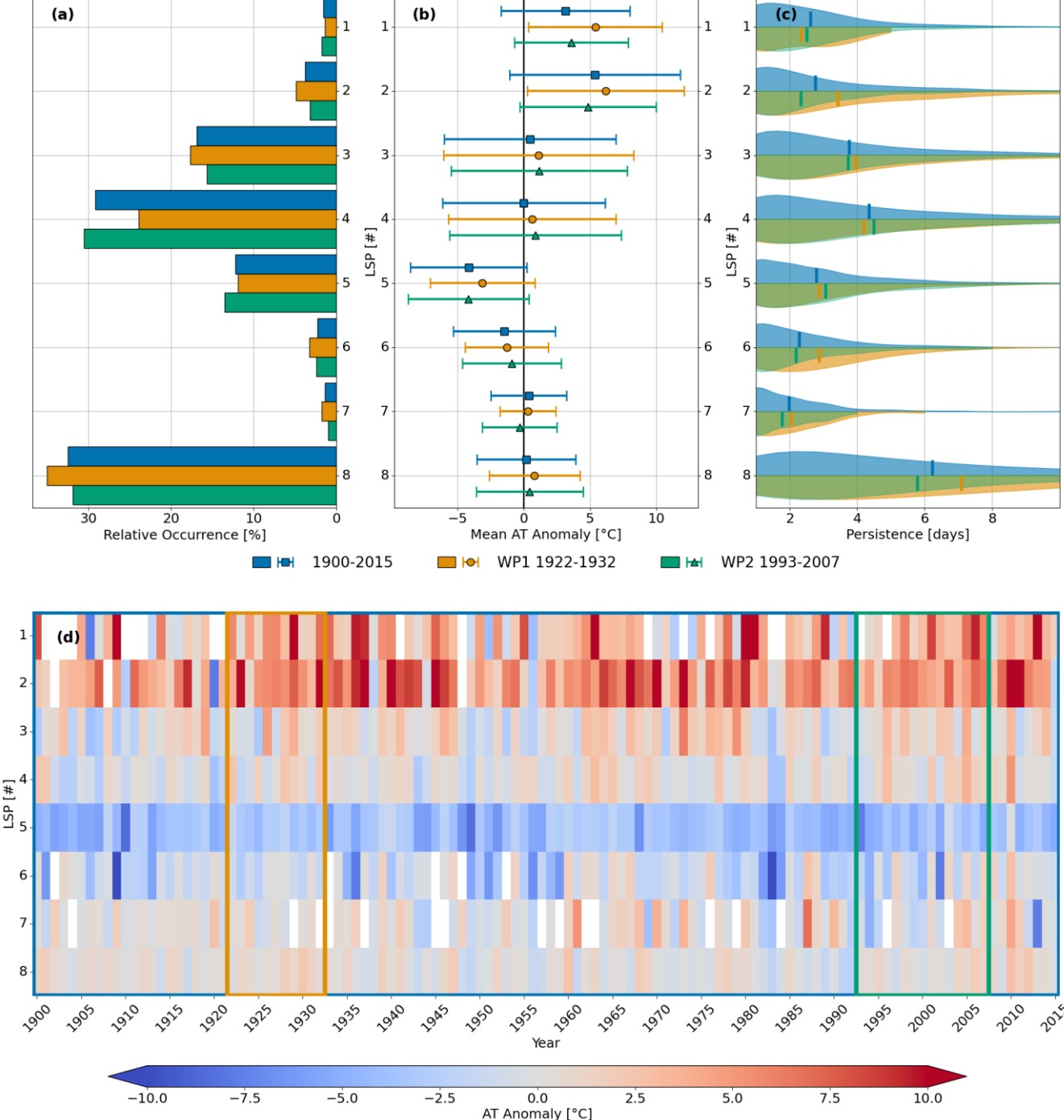

**Figure A1.4: Summary of the evaluation of the eight LSPs within the larger domain (120°W -20°E and 50-90°N) examined across the study periods. The study periods are color-coded throughout the plot as follows: WP1 (1922–1932) in orange, WP2 (1993–2007) in green, and the full period (1900–2015) in blue. (a) Distribution of relative occurrence of each LSP across the study periods. (b) Average AT anomaly per LSP, with markers indicating the mean anomaly and whiskers representing ±1 standard deviation. (c)**

Distribution of persistence in days per LSP, with bold lines indicating mean lengths. The full period is at the top and both WPs at the bottom. (d) Annual average AT anomaly per LSP, with coloured frames representing the study periods.

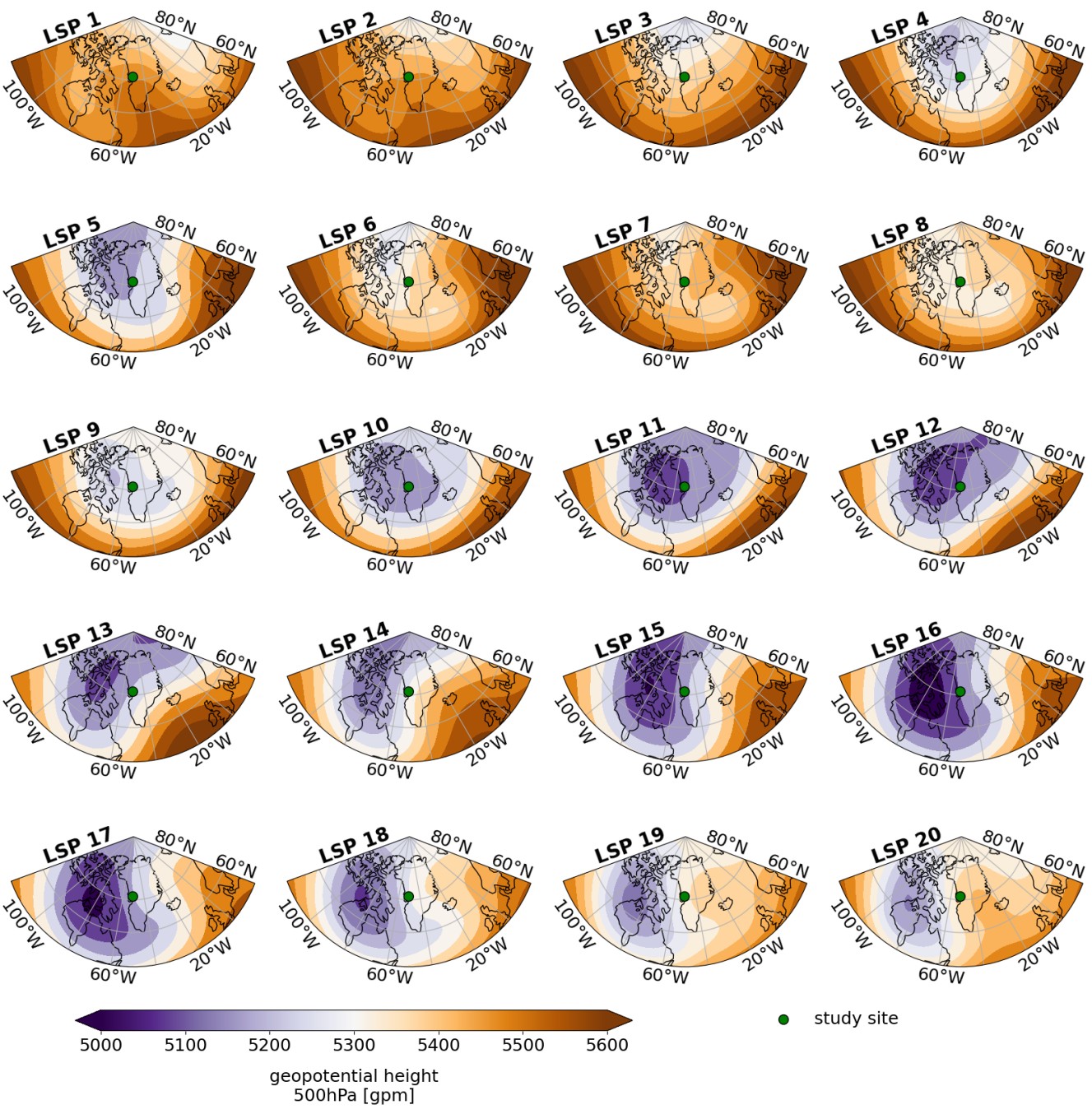

Figure A1.5: Geopotential height of the 500 hPa pressure level of the 20 LSPs as defined by SOM with a larger input domain of 120°W -20°E and 50-90°N. The study site is marked with a green dot. For visual clarity, the SOM patterns are displayed in a 2D

matrix; however, the underlying topology is one-dimensional, with neighbourhood relations applying only sequentially along a single line from the top left to the bottom right, i.e., following the numbering of the LSPs.

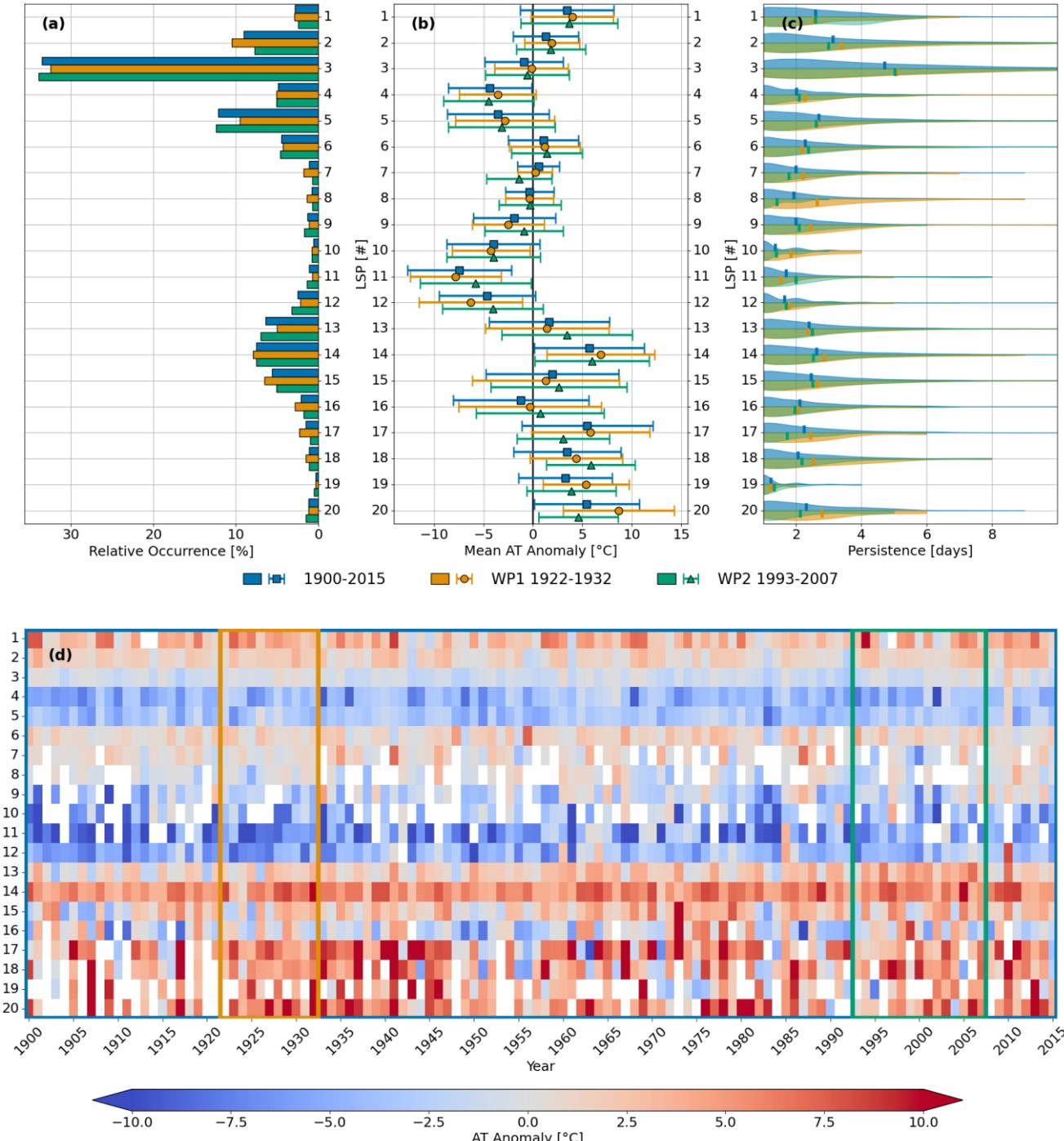

 Figure A1.6: Summary of the evaluation of the 20 LSPs with the larger domain (120°W -20°E and 50-90°N) examined across the study periods. The study periods are color-coded throughout the plot as follows: WP1 (1922–1932) in orange, WP2 (1993–2007) in

green, and the full period (1900–2015) in blue. (a) Distribution of relative occurrence of each LSP across the study periods. (b) Average AT anomaly per LSP, with markers indicating the mean anomaly and whiskers representing ±1 standard deviation. (c) Distribution of persistence in days per LSP, with bold lines indicating mean lengths. The full period is at the top and both WPs at the bottom. (d) Annual average AT anomaly per LSP, with coloured frames representing the study periods.

## A2. Seasonal annual AT anomaly

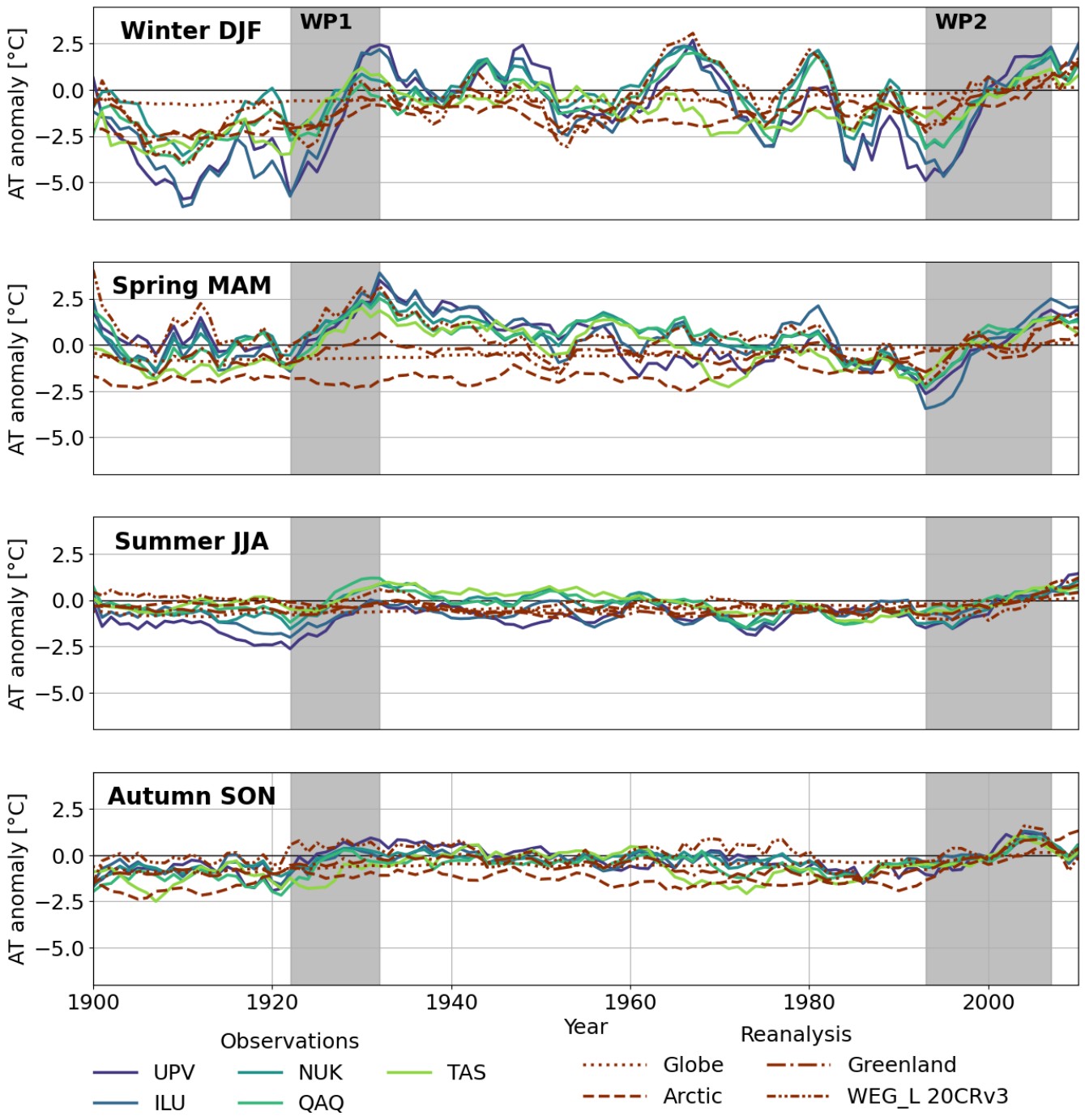

 **Figure A2.1. Seasonal annual AT anomaly with respect to reference period 1986-2015 at weather stations (Upernavik (UPV), Illulisaat (ILU), Nuuk (NUK), Qaqortoq (QAQ), Tasiilaq (TAS)), of 20CRv3 as spatial average of the Arctic, Greenland, globally and interpolated to the study site WEG_L, smoothed with a 5-year window rolling mean. The two defined WPs are marked with the grey background.**

**A3. Relative LSP occurrence**

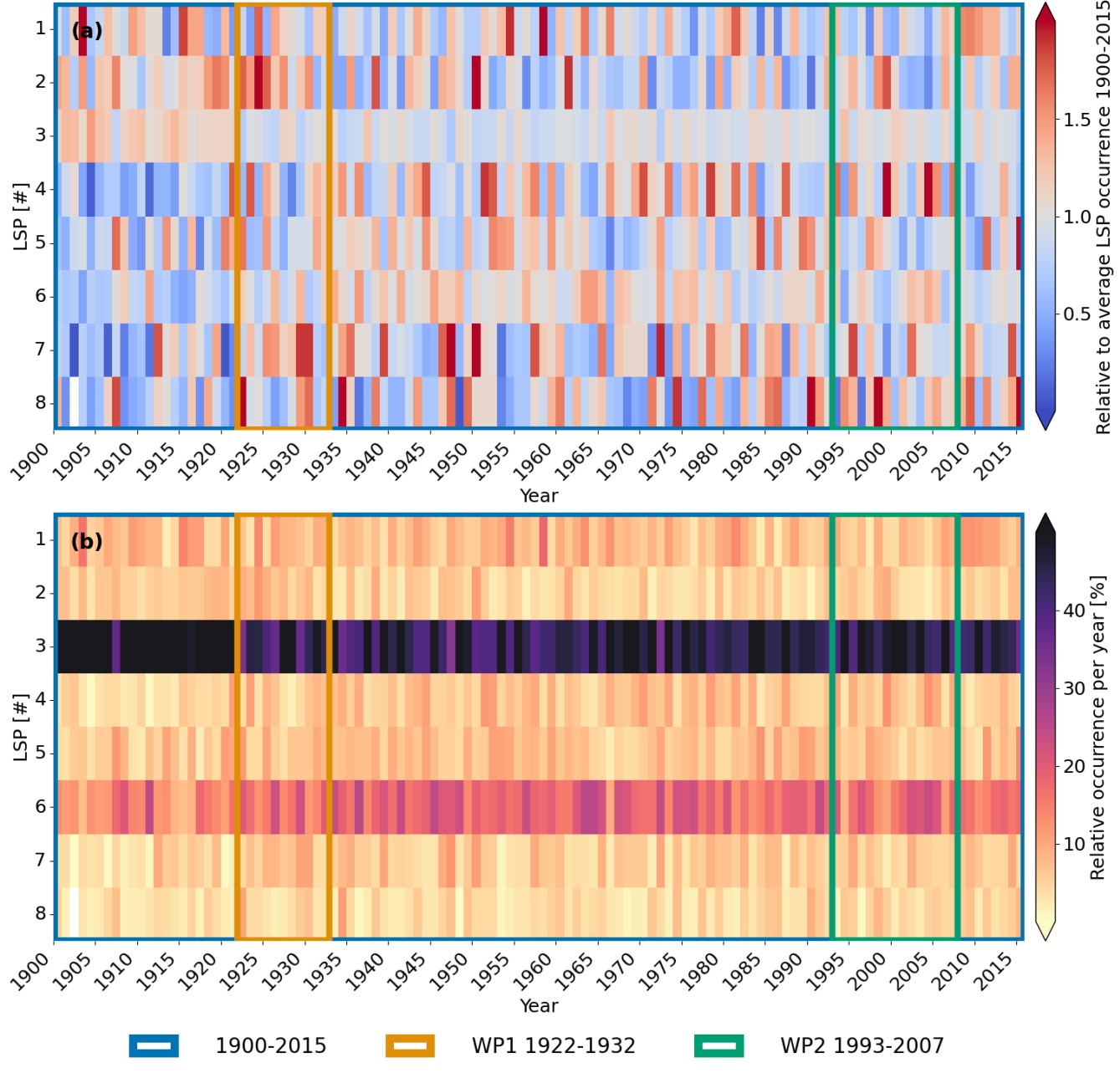

**Figure A3.1: The occurrence per LSP (a) relative to the average occurrence of a LSP in the full period 1900-2015 and (b) relative per year.**

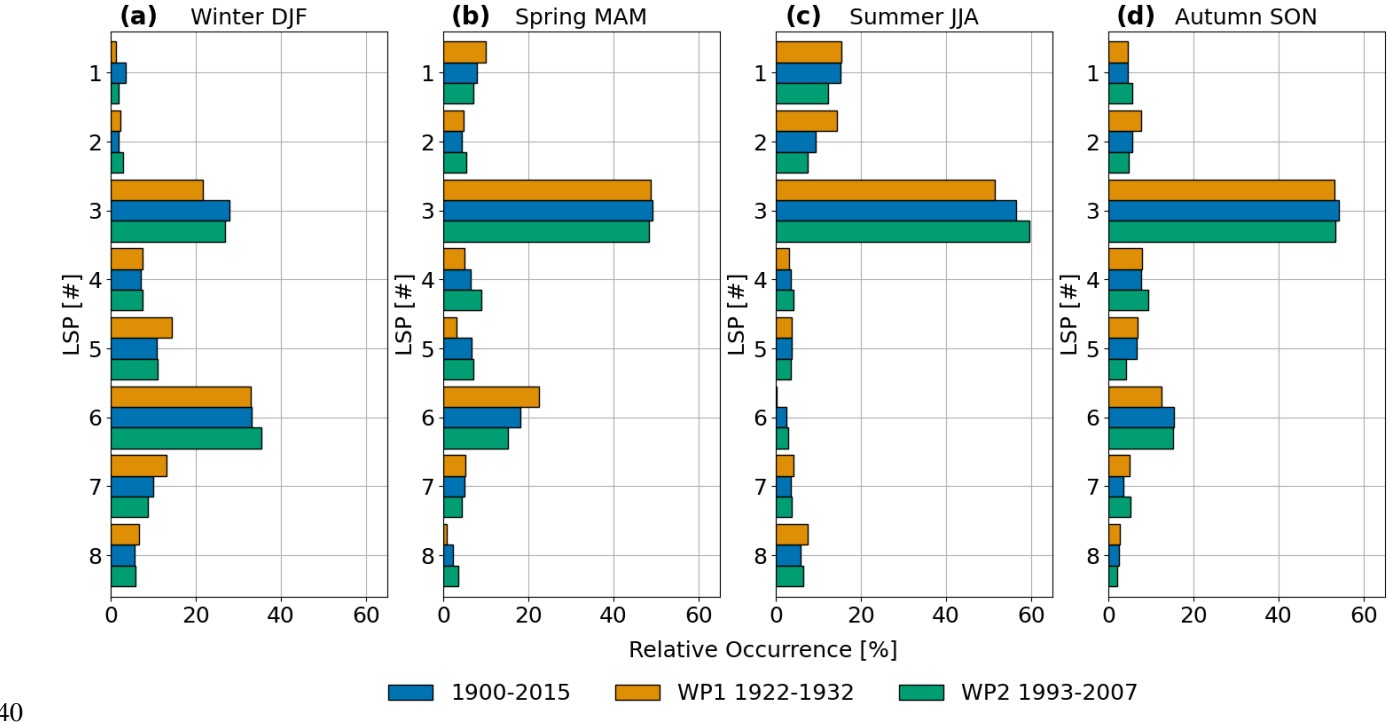

**Figure A3.2 Seasonal relative occurrence of the LSPs in the full study period and the WPs.**

## A4. Seasonal distribution of LSP on the 15 % warmest and coldest days in the WPs

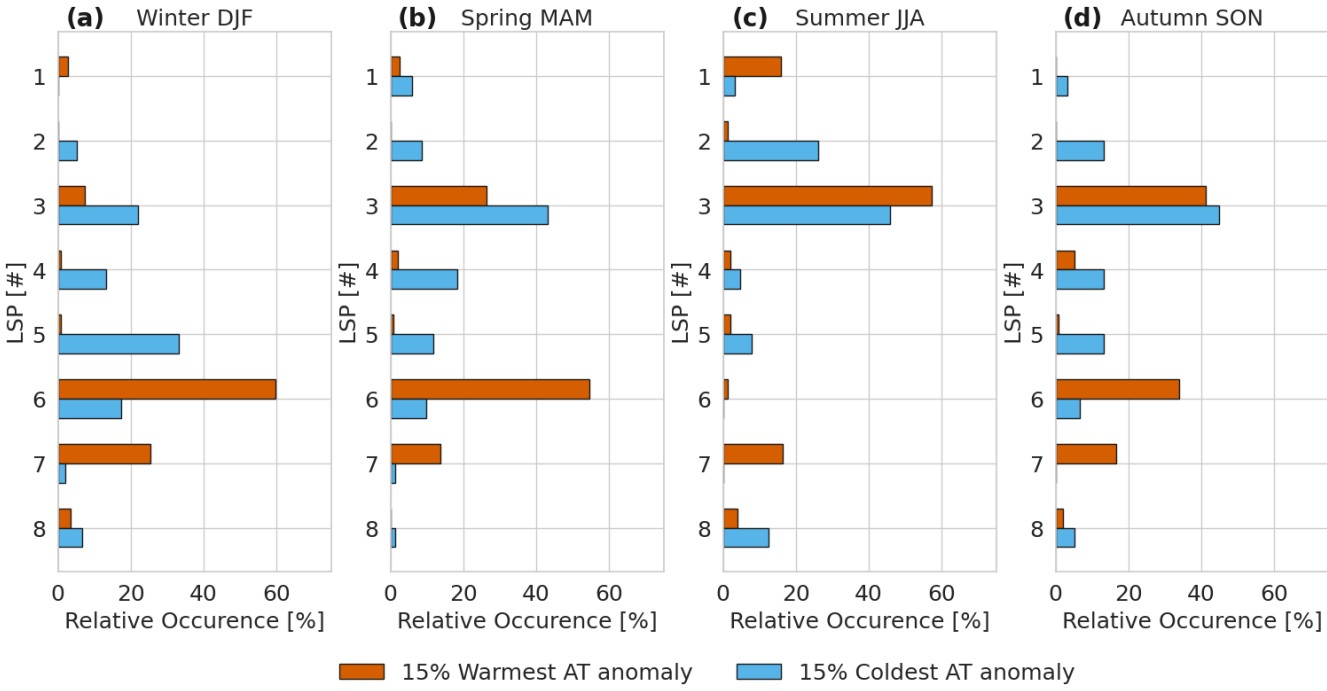

**Figure A4.1: Seasonal distribution of LSPs on the 15 % warmest and coldest days of WP1.**

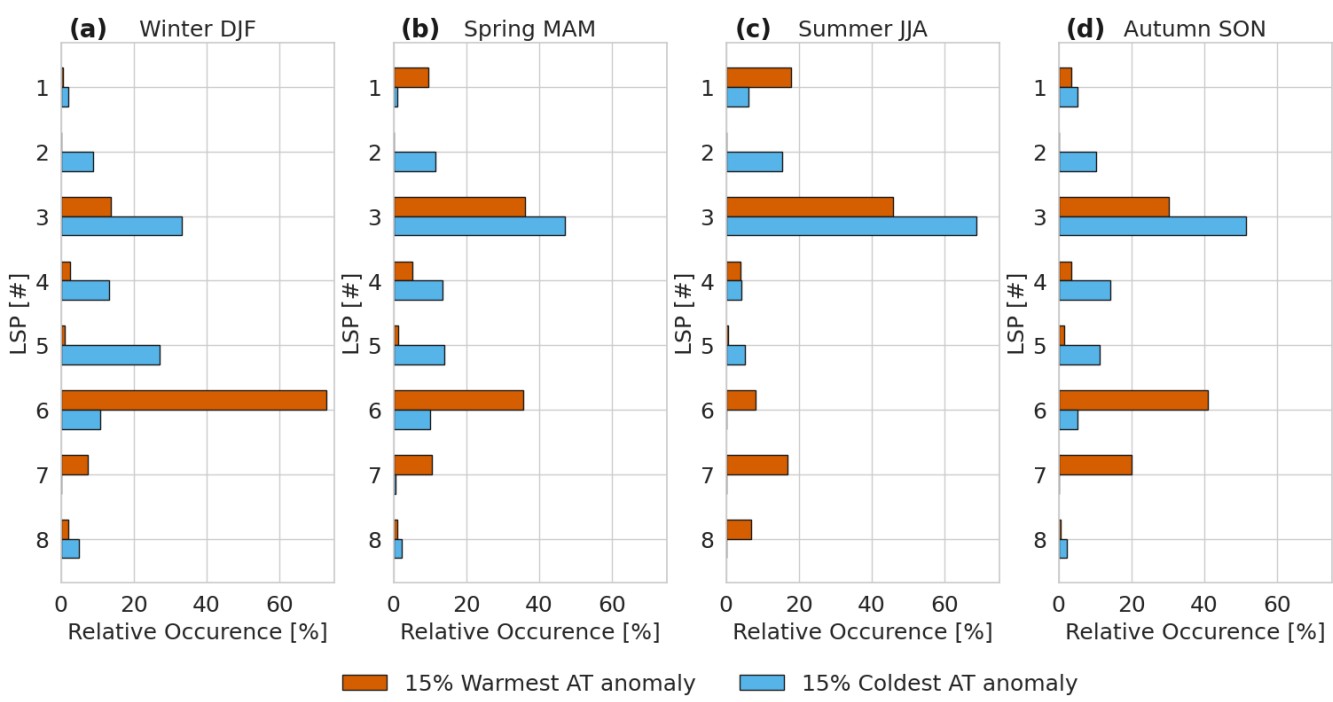

**Figure A4.2: Seasonal distribution of LSPs on the 15 % warmest and coldest days of WP2.**

**Code availability**

The code is available at the first authors

https://github.com/Florina3103/LSP_WP_analysis_git/tree/09fac112723281473c17db2e8dc46dff2a1960b6, and once the paper is accepted it will also be uploaded to zenodo under the reserved doi (10.5281/zenodo.14517648).

**Data availability**

The observations from DMI are available: https://opendatadocs.dmi.govcloud.dk/DMIOpenData
20CRv3 is available at NOAA PSL, Boulder, Colorado, USA, from their website at https://psl.noaa.gov/

**Supplement link:**

The link to the supplement will be included by Copernicus, if applicable.

**Author contribution:**

FRS was the main responsible for the analysis and the preparation of the manuscript. JAB supervised the project, SSC supported analysis and interpretation of the results, LHA gave input about the self-organizing map algorithm, all authors
contributed with reviewing the manuscript.

**Competing interests:**

The authors declare that they have no conflict of interest.

**Acknowledgements**

We thank NOAA for providing the 20Cv3 data and DMI for the observations. This research was funded by the Austrian
Science Fund (FWF) [P35388]. ChatGPT [https://chat.openai.com/] was used to draft single paragraphs of the manuscript for first ideas and improving language. We also thank Assoc. Prof. Doan for valuable support during the review process, as well as for insightful discussions regarding the methodology.

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
