# Peer review of "The role of atmospheric large-scale patterns for recent warming periods in Greenland from 1900-2015"

_EGUsphere, 2024_

## Referee Comment (RC1)

**Review of 'The role of atmospheric large-scale patterns for recent warming periods in Greenland'**

In this study, Schalamon et al. present a comprehensive analysis using a long-term observational dataset from Greenland combined with weather pattern clustering based on reanalysis data. The authors identify two distinct warming periods evident in observational records and supported by reanalysis data. The study uses Self-Organizing Maps (SOM) to investigate the role of changes in circulation patterns on warming periods and temperature anomalies. The study uses well established methods applied over centennial timescales, which sets it apart from other studies that typically focus on the satellite era. The questions that are addressed are relevant for the journal's scope and contribute to understanding the impact of changes in atmospheric circulation on Greenland climate and SMB. The figures in the manuscript are presented well and clearly. My comments are mostly around sensitivity of the results to choices in setting up the SOM. Furthermore, the manuscript could be strengthened by adding further interpretation of the results of the LSPs and their link with warming anomalies, and bringing these in context with known modes of variability and synoptic circulation features in the Arctic.

**Main comments**

- It is unclear from the introduction and methods why the WEG_L site is chosen as study area for the AT anomaly analysis. Further motivation is needed in the introduction or methods to clarify why the authors choose to link this analysis with the dataset described in Abermann et al. (2023).
- Are the SOM results dependent on seasonality in the geopotential height field? Previous studies (e.g., Cassano et al., 2015) suggest that using spatial anomalies instead of absolute fields can remove seasonal signals, focusing SOM analysis on gradients in geopotential height that drive advection.
- It is interesting that in LSP3 both positive and negative anomalies occur. Could the small number of clusters have resulted into several patterns being averaged into the zonal pattern shown in LSP3? Have the authors tested the results by training a larger SOM and seeing if those positive and negative anomalies still occur from a zonal pattern?
- It would be valuable to add whether warming trends in WP1 and WP2 show seasonal variation. Are certain seasons contributing more than others to the observed trends?
- I would suggest to add an additional figure in which the LSP occurrence per year is given for the full study period. This could show potential shifts in LSP occurrence in the cold period before the WP versus the warm period after that could explain the WP patterns.

- Is there seasonality in LSP occurrence over the full period? In that case I would suggest instead of showing relative changes during WPs, Fig. 5 could show absolute occurrences of LSPs across both the overall as well as the warming periods.
- Looking at the similarity in relative occurrence of the LSPs between the different periods I am surprised the distribution of LSPs is significantly different. Can the authors explain why a Chi-Square test is used and if this result would be robust with other significance tests?
- The manuscript could be strengthened by some further interpretation of the results from the SOM analysis and linking these with known modes of variability. For example, do the node occurrences correlate with the Arctic Oscillation or NAO? Could high occurence of LSP3 reflect conditions of strong polar vortex with less meandering, in which the Arctic and Greenland are often colder (agrees with Fig. 6. which is often during the positive phase of AO). During opposite conditions the polar vortex is weaker and weavier patterns form in the geopotential height field, which might explain the patterns detected by the other LSPs.
- The discussion could be strengthend by comparisons with studies linking extreme warming/melt events in Greenland to atmospheric conditions (e.g., Fettweis et. al 2013, Neff et. al, 2014, Hermann et. al, 2020)

**Specific comments**

Title: Consider including the study period to highlight the long temporal scope of the study.

L. 37: Without reading Abermann et al. 2023, it is unclear from this section what is meant with 'high-resolution observations'.

L. 65: add 'summer' after 'from the west'.

L. 80: Can you add more information on the weather station data, such as measured variables and presence of data gaps, or refer to the source for these details.

Sect. 4.1: The warming periods are based on significant warming trends in the weather station data. Are the trends in reanalysis over the same periods significant as well? For example, in Fig 2b doesn't look significant at the study site location.

Fig 2: Include the study site location also in Fig 2b.

L. 197: The Arctic doesn't seem to warm during this period, so rephrase to 'concentrated over Greenland' to also agree with your following statements.

L. 225: Can you clarify this statement? Often SOMs are shown as matrix in which neighboring nodes are most similar and further away nodes most different. In case of atmospheric circulation patterns neighboring nodes could be transitional from one synoptic state to another.

L. 315 – 320. This section would fit better in Methods section 2.2

L. 367: The difference in warming through LSP6 (dominantly in winter) and LSP3 (dominantly in summer) is interesting and could have more discussion here. What is the role of advection from continental vs oceanic regions? This could explain why in winter there is more warming during LSP 6 (relative warm ocean) and in summer warming from continental sources (LSP3).

**References**

Cassano EN, Glisan JM, Cassano JJ, Gutowski WJ Jr, Seefeldt MW (2015) Self-organizing map analysis of widespread temperature extremes in Alaska and Canada. Clim Res 62:199-218. https://doi.org/10.3354/cr01274

Fettweis, X., Hanna, E., Lang, C., Belleflamme, A., Erpicum, M., and Gallée, H.: Brief communication "Important role of the mid-tropospheric atmospheric circulation in the recent surface melt increase over the Greenland ice sheet", The Cryosphere, 7, 241–248, https://doi.org/10.5194/tc-7-241-2013, 2013.

Hermann, M., Papritz, L., and Wernli, H.: A Lagrangian analysis of the dynamical and thermodynamic drivers of large-scale Greenland melt events during 1979–2017, Weather Clim. Dynam., 1, 497–518, https://doi.org/10.5194/wcd-1-497-2020 , 2020.

Neff, W., G. P. Compo, F. M. Ralph, and M. D. Shupe (2014), Continental heat anomalies and the extreme melting of the Greenland ice surface in 2012 and 1889, J. Geophys. Res. Atmos., 119, 6520–6536, doi:10.1002/2014JD021470.

---

## Author Response (AR1)

Dear editor and reviewers,

We are very grateful for your very constructive review and appreciate the valuable time put into this. We went carefully through your suggestions and answered everything below. We incorporated changes into the manuscript where we believe that the adjustments will improve the manuscript.

Below, we present the review comments in **bold**, followed by our responses in *italic*. Any amendments we made to the revised manuscript are highlighted in green. The line numbers indicated with [l.XXX] are based on the revised manuscript.

We hope that the changes will convince both the reviewers and the editor and are looking forward to a decision.

Once again, many thanks for the valuable input and all the best – on behalf of the author team,

Florina Schalamon

**Answer review comment #1**

**Main comments**

**- It is unclear from the introduction and methods why the WEG_L site is chosen as study area for the AT anomaly analysis. Further motivation is needed in the introduction or methods to clarify why the authors choose to link this analysis with the dataset described in Abermann et al. (2023).**

*Thank you for this important comment to the need to explain the motivation for selection of study site. This study is part of the WEG_Re project, which investigates the influence of climate drivers on glacier changes since Alfred Wegener's last expedition to Greenland (1930/31). WEG_L is the position of an automated weather station implemented within WEG_Re, that coincides with the location of a weather station from Wegener's historic expedition.*
*The submitted article should be seen in the context of a number of studies such as Abermann et al. (2023) who give a general background of environmental changes, or Scher et al. (submitted), who demonstrate that even short observational periods can improve uncertainty estimates in reanalysis data.*
*Beyond project logics, our study also covers a scientifically relevant region where strong AT trends are present (see Fig. 2 (b) and (c)) in both WPs.*
*To address the reviewer's concern that this connection is unclear, we added additional explanation in the revised manuscript.*

Using the same location as Wegener's expedition allows for a direct comparison between past and present atmospheric conditions, enabling a better understanding of long-term changes and the role of large-scale atmospheric patterns (LSPs) in shaping regional climate variability.  [l.43]

From a climatological perspective, WEG_L is located in a region where AT changes are among the strongest during both WPs, as seen in Fig. 2 (b) and (c). The particular strong and significant warming in this area further supports WEG_L as a representative study site, making it a well-suited choice for this study. [l.230]

**- Are the SOM results dependent on seasonality in the geopotential height field? Previous studies (e.g., Cassano et al., 2015) suggest that using spatial anomalies instead of absolute fields can remove seasonal signals, focusing SOM analysis on gradients in geopotential height that drive advection.**

*Thanks, this touches an interesting point. It would indeed be possible to compute the SOMs on spatial anomalies instead of absolute fields. If the anomalies were computed on a non-seasonal basis, then this would lead to exactly the same results, as differences would cancel out in the distance functions. If the anomalies were computed on a seasonal basis (e.g., with a running seasonality), then the results would potentially be different. While this would be a valid alternative approach for this type of analysis, this would answer a slightly different research question – namely "which large-scale anomalies drive regional warming?", in contrast to our analysis, which answers one of the research questions in our paper, namely "which large scale patterns drive regional warming?". In order to keep conciseness, we suggest keeping the focus with absolute fields. We address this point in the revised manuscript's discussion, adding:*

[…] from 1900 to 2015. While alternative approaches using pressure anomalies could offer different insights into seasonality, we base our SOM analysis on absolute pressure fields to directly assess the influence of LSPs on regional warming. [l.157]

**- It is interesting that in LSP3 both positive and negative anomalies occur. Could the small number of clusters have resulted into several patterns being averaged into the zonal pattern shown in LSP3? Have the authors tested the results by training a larger SOM and seeing if those positive and negative anomalies still occur from a zonal pattern?**

*This is a valid and interesting point. Given the high relative occurrence of LSP3 and its broad range of associated anomalies, it is likely that different conditions are grouped within this pattern. This is expected climatologically, as zonal airflow can result from various large-scale pressure configurations. Additionally, the SOM algorithm assigns more cluster centres where data density is higher, meaning that LSP3, containing a large number of days, likely represents multiple similar patterns in a less dense input space.*
*While increasing the number of SOM clusters could, in principle, help differentiate these patterns, it introduces challenges. Some LSPs would become too rare creating special cases rather than offering a streamlined interpretation of dominant large-scale patterns. Any SOM analysis involves some arbitrariness in parameter selection, including the choice of cluster numbers. Our selection aligns with previous studies (Preece et al., 2022; Schmid et al., 2023; Schuenemann & Cassano, 2009), ensuring comparability. As noted in l. 158, we aimed to avoid manually regrouping clusters after the analysis with SOM and opted for a manageable number of clusters.*

*We also tested configurations with more (9,10,15,20,25,30) and fewer (6,7) clusters. While this naturally altered the distribution, it did not improve interpretability. Specifically, increasing the number of clusters did not lead to a meaningful differentiation of AT anomalies within zonal LSPs. It needed 30 clusters to split the zonal flow LSP into two clusters and both showed similar AT anomalies associated with it.*
*Our goal was to maintain reproducibility, minimize manual intervention, and keep the number of patterns manageable. We found that our chosen number of clusters (8) provided a good balance.*

*We included the example of 20 cluster centres in the appendix A1 of the revised manuscript, as well as the following sentence to address this point.*

We performed test runs for different number of clusters (see appendix A1), but  we aimed to avoid the need for subgrouping after applying the SOM algorithm, opting instead for a straightforward and consistent clustering approach. [l. 176]

In the following we show different parameters we tested for the SOM method. We tested different number of nodes – 6,7,8,9,10,15,20,25,30. As an example we added here the results with 20 nodes. As we wanted to avoid the manual regrouping after SOM as done by Schmidt et al. (2023) and Schuenemann and Cassano (2009) we concluded that eight clusters are 485 sufficient to show the expected large-scale patterns over Greenland and that they show the expected warm and cold patterns. More clusters do not result in more details. We further found that the key results are not sensitive to the number of clusters. [l.468]

**- It would be valuable to add whether warming trends in WP1 and WP2 show seasonal variation. Are certain seasons contributing more than others to the observed trends?**

*Thank you for your helpful comment. We have indeed examined the seasonality of warming trends and will include this aspect in the revised manuscript. Figure R1 shows that winter AT trends are larger than in the other seasons in both WPs, highlighting the dominant role of winter variability. This finding aligns with Box et al. (2009), who demonstrated that positive annual temperature anomalies over Greenland are largely driven by winter temperature variations. To address this, we added the following sentences in the introduction and the results part, as well as Fig. R1 as Fig. A2.1 in the appendix:*

[...] from 1994 to 2007. Their study shows that these warming trends are not uniform across seasons, with winter temperatures exhibiting much greater variability than summer temperatures. [l.28]

The seasonal analysis confirms the findings from Box et al. (2009), that the AT increase is strongest in winter (see Fig. A2.1), while the smaller anomalies are observed in summer and autumn. [l.233]

[Figure]

Figure R1: Seasonal annual AT anomaly with respect to the reference period 1986-2015 at weather stations (Upernavik (UPV), Illulisaat (ILU), Nuuk (NUK), Qaqortoq (QAQ), Tasiilaq (TAS)) and in addition the 20CRv3 as spatial AT anomaly average of the Arctic, Greenland, globally and interpolated to the study site WEG_L, smoothed with a 5-year window rolling mean. The two defined WPs are marked with the grey background. (This is Fig. A2.1 in the revised manuscript.)

**- I would suggest to add an additional figure in which the LSP occurrence per year is given for the full study period. This could show potential shifts in LSP occurrence in the cold period before the WP versus the warm period after that could explain the WP patterns.**

*Thanks, this is of course another useful possibility to display the LSP occurrence per year and we investigated this aspect beforehand. Figure R2 (a) shows the relative occurrence per LSP over the years scaled to the average occurrence over the full period 1900-2015. A value above/below 1 means a higher/lower occurrence of that pattern compared to the average occurrence of that pattern. Figure R2 (b) shows the course of the relative occurrence of one specific LSP per year over the full study period.*

*Despite it giving an additional comprehensive perspective, we did not find a clear trend of specific LSPs over the whole study period. For instance, while LSP2 (a cold pattern) occurs more frequently than average during WP1, LSP7 also shows an increased occurrence, making it difficult to establish a direct link.*

*We aimed to observe changes as similar trends have been identified in other Arctic regions, such as the reported rise in cyclonic activity by Hanssen-Bauer et al. (2019) and Wickström et al. (2020).*

*We included the shown figure in the appendix A3 of the revised manuscript to complete the analysis of the LSP occurrence. Additionally, we will include the following sentences.*

[...] significant difference robust. We did not find clear evidence of trends in LSP distribution per year. Figure A3.1 in the appendix provides further details, illustrating the annual relative occurrence of each LSP compared to its average occurrence over the full period (1900–2015) (a), as well as the relative occurrence per year (b). [l.272]

[Figure]

Figure R2:(a) the relative occurrence per LSP over the years scaled to the average occurrence over the full period 1900-2015. A value above/below 1 means a higher/lower occurrence of that pattern compared to the average occurrence of that pattern. (b)

shows the course of the relative occurrence of one specific LSP per year over the full study period. (This is Fig. A3.1 in the revised manuscript.)

**- Is there seasonality in LSP occurrence over the full period? In that case I would suggest instead of showing relative changes during WPs, Fig. 5 could show absolute occurrences of LSPs across both the overall as well as the warming periods.**

*We appreciate this point and show the relative occurrence of LSPs during WPs and FPs in Fig. R3. Our point on small but significant differences between the WPs and the full study period is further highlighted in Fig. 5, where we show it in relative terms – in principle containing the same information. To address the reviewer's point, we included Fig. R3 as Fig. A3.2 in the appendix A3 and add the following explanation to the revised manuscript:*

In a first step the relative occurrences were analysed seasonally ==and the full overview is displayed in Fig. A3.2. To highlight the small but significant differences between the occurrence of the LSPs, Fig.5 shows the occurrence in the WPs== relative to the occurrence in the entire study period 1900-2015. [l.307]

[Figure]

Figure R3: Seasonal relative occurrence of the LSPs in the WPs and in the entire study period 1900-2015. (This is Fig. A3.2 in the revised manuscript.)

**- Looking at the similarity in relative occurrence of the LSPs between the different periods I am surprised the distribution of LSPs is significantly different. Can the authors explain why a Chi-Square test is used and if this result would be robust with other significance tests?**

*Indeed, subtle differences must be confirmed with appropriate statistical testing. To compare the occurrence of LSPs between the full study period and the WPs, we tested whether there was a significant difference between the expected occurrence (based on the full period) and*

*the observed occurrence within each WP. We acknowledge that the differences in Fig. 4(a) appear small, which is why we highlighted the differences in Fig. 5. To ensure the robustness of our results, we repeated the significance test 1500 times.*

*The Chi-Square test is appropriate for comparing observed frequencies of categorical data in a contingency table against expected frequencies under the null hypothesis of no association between the two distributions. In our case, it assesses whether the observed frequency of LSPs in a WP deviates significantly from the expected frequency derived from the full study period. This test evaluates the overall difference in LSP distribution rather than individual deviations for specific LSPs.*

*To validate our results, we performed additional analyses with other significance tests for the independence of two categorical distributions. For this we performed the G-Test (log-likelihood ratio test) and Fisher's exact test. Additionally, we also applied the permutation-based and Monte Carlo methods as variation of the standard Chi-Square test. These variations account for small frequency of one category in the contingency table, which could cause the default asymptotic approximation of the standard Chi-Square test to be inaccurate.*

Table R1: P-values from different significance tests comparing the periods, including the Chi-Square test, G-Test, Fisher's Exact Test, and permutation-based and Monte Carlo methods.

| Method | WP1 vs WP2 | WP1 vs full period | WP2 vs full period |
|---|---|---|---|
| *Chi Square* | *2.8e-07* | *6.2e-08* | *0.00097* |
| *G-test* | *3e-07* | *2.08e-07* | *0.0011* |
| *Montecarlo variation* | *0.0001* | *0.0001* | *0.0015* |
| *Permutation variation* | *0.0001* | *0.0001* | *0.0011* |
| *Fisher Exact* | *0.0001* | *0.0001* | *0.0013* |

*Tab. R1 gives an overview of the p-values obtained with the different tests and for the different periods that have been compared categorically. It shows that there is a significant difference between the LSP distribution in the WPs and the full period (threshold alpha = 0.05) for all tests and all periods.*

*We added the following statement to the manuscript as an explanation for choosing the chi-square test:*

[...] a Chi-square test was performed. The Chi-Square test is appropriate for comparing observed frequencies of categorical data in a contingency table against expected frequencies under the null hypothesis of no association between the two distributions. That means in our case that the observed frequency of LSPs in one WP is compared to the expected frequency of the LSPs in the full study period. [l.191]

*We also included a sentence about the additional test results along:*

However, statistical testing using a Chi-Square Test reveals a significant difference in the distribution of LSPs among WP1, WP2 and the full study period. This result is also supported by various other significance tests, including Fisher's Exact Test and the G-Test, as well as alternative approaches to the standard Chi-Square test, such as the Monte Carlo and permutation-based resampling methods. [l.269]

**- The manuscript could be strengthened by some further interpretation of the results from the SOM analysis and linking these with known modes of variability. For example, do the node occurrences correlate with the Arctic Oscillation or NAO? Could high occurrence of LSP3 reflect conditions of strong polar vortex with less meandering, in which the Arctic and Greenland are often colder (agrees with Fig. 6. which is often during the positive phase of AO). During opposite conditions the polar vortex is weaker and weavier patterns form in the geopotential height field, which might explain the patterns detected by the other LSPs.**

*Thank you for your insightful suggestion and for highlighting the potential connections between the LSPs and large-scale climate modes like the Arctic Oscillation (AO). We agree that the high occurrences of LSP3 could reflect conditions associated with a strong polar vortex, where a more zonal atmospheric circulation leads to colder conditions over Greenland, especially during the positive phase of the AO. This phase is characterized by a more stable and confined polar vortex, which typically results in less meandering of the jet stream and colder temperatures across the Arctic. Conversely, during the negative phase of the AO, the polar vortex weakens, allowing for increased meandering of the jet stream and the formation of more variable weather patterns, which could align with the LSPs observed in other periods.*

*While these connections are well-documented in the literature, our study primarily focuses on the variability of the distribution of LSPs during the warming periods (WPs), rather than exploring the mechanisms driving these patterns. The aim was to assess how different atmospheric circulation patterns contribute to temperature anomalies during these specific periods of warming, and linking LSP occurrences directly to indices such as the AO or NAO would require a more detailed analysis of these dynamics, which goes beyond the scope of the current manuscript. However, we do recognize the importance of these climate modes in shaping atmospheric conditions over Greenland, and this will certainly be an important consideration for future research as we further explore the role of large-scale circulation patterns in driving temperature variability.*

*In the revised manuscript we added the following statement to strengthen the interpretation as the reviewer suggested.*

[...] amplify warming trends. ==The variability of LSP occurrences may also reflect broader atmospheric circulation changes, potentially influenced by climate modes such as the Arctic Oscillation (AO). A strong polar vortex, often associated with a positive AO phase, can contribute to more stable, zonal airflow patterns, while a weaker vortex during a negative AO phase allows for increased meandering and variability in geopotential height fields. These large-scale influences can provide additional context for understanding the observed shifts in LSP frequency and their role in shaping local AT anomalies.== [l.376]

**- The discussion could be strengthend by comparisons with studies linking extreme warming/melt events in Greenland to atmospheric conditions (e.g., Fettweis et. al 2013, Neff et. al, 2014, Hermann et. al, 2020)**

*Thank you for this valuable suggestion, which coincides with our research agenda very nicely. In the next phase of our project, we will investigate the historical and modern datasets at the study site, explicitly linking atmospheric conditions to recorded melt events. As part of this analysis, we will assess the role of the LSPs defined in this study as potential drivers of these*

*events. Since this aspect is part of a new manuscript we are working on, we have not included it in this. However, we added a sentence outlining this future research direction in the conclusion in the revised manuscript.*

[...] under different warming scenarios. A further application of our research will be to connect LSPs to measured and modelled surface mass balance of the glacier in the study area, as it is known that extreme melt events are closely linked to atmospheric circulations (Fettweis et al., 2013; Hermann et al., 2020; Neff et al., 2014). [l.459]

**Specific comments**

**Title: Consider including the study period to highlight the long temporal scope of the study.**

*We changed the title to: "The role of atmospheric large-scale patterns for recent warming periods in Greenland from 1900-2015".*

**L. 37: Without reading Abermann et al. 2023, it is unclear from this section what is meant with 'high-resolution observations'.**

*We rephrased the expression to "Observations with high temporal and spatial resolution".* [l.40]

**L. 65: add 'summer' after 'from the west'.**

*This was added in the revised manuscript.* [l.80]

**L. 80: Can you add more information on the weather station data, such as measured variables and presence of data gaps, or refer to the source for these details.**

*Cappelen et al., (2021) gives the full background to the DMI Historical climate data collection used for this study. This reference is already included in the description of the weather stations in l.96. We added the following sentence for clarification in the revised manuscript.*

[...] station on the east coast. More detailed information can be found in Cappelen et al. (2021). [l.97]

**Sect. 4.1: The warming periods are based on significant warming trends in the weather station data. Are the trends in reanalysis over the same periods significant as well? For example, in Fig 2b doesn't look significant at the study site location.**

*Thank you for the valuable suggestion, which led us to refine the representation of the data. To ensure consistency, we applied the smoothed AT anomaly trends in all panels of the figure (see Fig. R4), providing a clearer depiction of the significant warming at the study site. Additionally, we verified all smoothed AT anomaly trends (Fig. 2a) during the WPs at the observation stations and in the reanalysis data for WEG_L, confirming that the trend remains statistically significant.*

*In the revised manuscript, we updated Fig. 2 with the following figure:*

[Figure]

Figure R4: (a): Annual AT anomaly with respect to reference period 1986-2015 at weather stations (Upernavik (UPV), Illulisaat (ILU), Nuuk (NUK), Qaqortoq (QAQ), Tasiilaq (TAS)), of 20CRv3 as spatial average of the Arctic, Greenland, globally and interpolated to the study site WEG_L, smoothed with a 5-year window rolling mean. The two defined WPs are marked with the grey background. (b) and (c): spatial representation of the Sen's slope estimator for the WPs. The colour shows the AT trend for one grid cell of 20CRv3, the "." hashing indicates grid cells where the trend is significant (Mann-Kendall test). (This figure is the updated Fig.2 in the revised manuscript.)

**Fig 2: Include the study site location also in Fig 2b.**

*This is adjusted as we include the Fig. R4 above in the revised manuscript as the updated Fig.2.*

**L. 197: The Arctic doesn't seem to warm during this period, so rephrase to 'concentrated over Greenland' to also agree with your following statements.**

*This is corrected in the revised manuscript.* [l.218]

**L. 225: Can you clarify this statement? Often SOMs are shown as matrix in which neighboring nodes are most similar and further away nodes most different. In case of**

**atmospheric circulation patterns neighboring nodes could be transitional from one synoptic state to another.**

*This statement was meant for the naming convention. That LSP 1 is the first does not have any meaning as it is just the first cluster centre found during the training part of SOM with choosing random input. We rephrased the sentence into:*

Note that the numbering of LSPs (e.g., LSP 1, LSP 2, …) is arbitrary, as it is based on the first cluster identified during the SOM training process, which depends on randomly chosen initial input data. [l.252]

**L. 315 – 320. This section would fit better in Methods section 2.2**

*We moved l. 317-320 to the end of the suggested section. So that the decision to choose 20CRv3 based on the prior study is in the method part with minor adjustments to embed it in the text. See the following:*

The approach follows the method described in Abermann et al. (2023). They evaluated the performance of the 20CRv3 and CERA-20C reanalysis models for two non-assimilated stations within the study area. Both models were interpolated and adjusted to the station altitudes, and their findings indicate that 20CRv3 aligns more closely with observations from 1930 and 1931 than CERA-20C. For this reason, we selected 20CRv3 for our analysis. Slivinski et al. (2021) confirmed the general good agreement between 20CRv3 and measured temperature. [l.119]

*We left the comparison of the reanalysis at WEG_L and the observations at the weather stations in the discussion part, as this is discussed further in the following paragraph.*

**L. 367: The difference in warming through LSP6 (dominantly in winter) and LSP3 (dominantly in summer) is interesting and could have more discussion here. What is the role of advection from continental vs oceanic regions? This could explain why in winter there is more warming during LSP 6 (relative warm ocean) and in summer warming from continental sources (LSP3).**

*Thanks for the input. We added the following sentences to include your argument.*

[...] the effect on the local AT anomaly. Further the difference in warming between LSP6 (dominantly in winter) and LSP3 (dominantly in summer) can likely be attributed to the source of advection. During winter, LSP6 is associated with advection from warmer oceanic regions, while LSP3 in summer is influenced by advection from continental sources, which explains the seasonal variation in warming. [l.408]

**Answer review comment #2**

**Major questions:**

**- Large-scale, low-frequency patterns of atmospheric circulation will change over the seasons. Did the authors consider conducting the SOM analysis on a seasonal basis? While the seasonal analysis in Figure 5 is helpful, I question whether combining all the data to calculate the SOMs would provide the same patterns as SOMs conducted for each season.**

*Thank you for your thoughtful comment. We agree that large-scale circulation patterns can vary by season, and conducting seasonal SOM analyses could indeed reveal more distinct seasonal structures. Our research question focuses on identifying dominant large-scale circulation patterns associated with regional warming during the defined warming periods, independent of season as an initial step. The seasonal analysis shown in Fig. 5 already indicates that the identified LSPs occur across multiple seasons. Therefore, while seasonal SOMs could yield additional insights, they would address a different question, and we consider the aggregated SOM approach based on absolute fields appropriate for the scope of this study. Nonetheless this is an interesting point and could be a next step in the future to expand this analysis.*

**- The authors should further explain the interest in the WEG_L station at Qaamarujup Sermia. It is unclear why they selected this station and how it was useful to the analysis.**

*Thank you for this important comment and for highlighting the need to clarify the rationale behind the selection of WEG_L as the study site. As already noted in response to a related reviewer comment, WEG_L is part of the WEG_Re project, which investigates the influence of atmospheric drivers on glacier change since Alfred Wegener's 1930/31 expedition to Greenland. The site hosts an automated weather station installed at the same location as one of Wegener's original observation points, offering a rare opportunity to explore long-term atmospheric changes with historical context.*

*Scientifically, WEG_L lies in a region that exhibits particularly strong and significant air temperature (AT) trends during both defined warming periods (WPs), as shown in Fig. 2 (b) and (c). This makes it not only a logistically justified site within the project context but also a climatologically meaningful choice for studying the influence of large-scale circulation patterns (LSPs) on regional AT anomalies.*

*To address the reviewer's concern that this connection is unclear, we added following explanation to the revised manuscript.*

Using the same location as Wegener's expedition allows for a direct comparison between past and present atmospheric conditions, enabling a better understanding of long-term changes and the role of large-scale atmospheric patterns (LSPs) in shaping regional climate variability.  [l.43]

From a climatological perspective, WEG_L is located in a region where AT changes are among the strongest during both WPs, as seen in Fig. 2 (b) and (c). The particular strong and significant warming in this area further supports WEG_L as a representative study site, making it a well-suited choice for this study. [l.230]

**- What is the value in looking at "warming periods", periods of increasing temperature (i.e., first derivative of temperature), rather than warm periods (some threshold value above a long-term mean)? I would imagine the circulation patterns are more clearly associated with "warm periods" than "warming periods".**

> **- The authors mention an "additional approach" looking at the warmest and coldest days (l. 179-180). Should this be more prominent and mentioned earlier in the manuscript?**

*Thank you for raising this point. We chose to focus on warming periods—defined as periods of increasing temperature—rather than warm periods (i.e., days exceeding a fixed threshold) because our aim was to investigate whether and how large-scale circulation patterns contribute to the change in temperature. Warm periods are typically defined relative to a climatological baseline, depending on that they change and are more suited to characterizing extremes or absolute temperature levels. In contrast, what we define as 'warming periods', is independent of the reference baseline, which allows us to assess the role of circulation in shaping long-term temperature trends, central to our research question.*

*While both approaches are valuable, they answer different questions. To complement this trend-based approach, we included an additional analysis of the warmest and coldest days, which offers insight into the circulation patterns associated with extreme temperature events. We agree with the reviewer that this part should be mentioned earlier in the manuscript and will adjust the introduction accordingly to better reflect its relevance.*

**- The spatial pattern of the air temperature trend that reaches west Greenland appears to extend from Baffin Bay (Figs. 2b and c; also see final minor question). Have the authors considered looking at trends in SSTs or sea ice coverage?**

*Thank you for this thoughtful suggestion. We agree that trends in SSTs and sea ice coverage in Baffin Bay are highly relevant, especially given the spatial pattern of warming extending into West Greenland. While we did not include SST or sea ice trends in the present analysis, our focus was on assessing how existing large-scale circulation patterns (LSPs) affect local air temperature (AT) at the study site, rather than investigating the drivers behind the occurrence or evolution of these LSPs themselves.*

*Long-term, continuous and homogenized records of sea ice coverage for the full study period (1900–2015) are limited, which constrains their direct integration into this analysis. However, we fully agree that case studies investigating the interaction between sea ice, SSTs, and specific LSPs—particularly during strong warming phases—represent an important avenue for future work.*

**- Previous work has used Empirical Orthogonal Functions (or Principal Components Analysis) to identify patterns in low-frequency modes of atmospheric circulation, including identifying NAO, using 700hPa or 500hPa heights. One of the key steps in such work is understanding the physical meaning of the patterns identified. What advantages are there to using the SOM analysis relative to EOFs? Does each pattern represent a physically meaningful mode of circulation?**

*EOFs are indeed widely used to identify patterns in atmospheric circulation. EOFs are based on maximizing variance, and when combined with a clustering algorithm – e.g. K-means – it can be used to cluster circulation patterns. SOMs can be used for the same purpose – clustering circulation patterns. The advantage is that they are slightly more generic and need less assumptions than EOFs. Based on this, and the wide use in recent studies (e.g. Hartl et al., 2020, 2023; Hofsteenge et al., 2024; Mattingly et al., 2018; Mioduszewski et al., 2016; Preece et al., 2022; Schmidt et al., 2023; Schuenemann & Cassano, 2009), we opted for SOMs in this study. We also agree that understanding the physical meaning of the identified patterns is crucial. Note that neither EOFs nor SOM utilizes any physics information, and thus neither is guaranteed to give physically meaningful groupings of similar conditions. For both methods, an explanation based on meteorological expertise is necessary. This is what we have done in section 4.2.*

**Minor questions:**

1. **30: Reducing snow cover on the ice but also physically changing the structure of snow and firn, the key is reducing the albedo.**

   *Yes, this is a more holistic point to changes in albedo. We rephrased to:*

   Changes in AT can influence Greenland's ice dynamics through feedback mechanisms linked to surface albedo. Rising temperatures lead to reduced snow and ice cover, increased exposure of bare ice and land, the formation of melt ponds, and progressive darkening of the snow and ice surface due to melting and the accumulation of impurities. These processes all lead to a   lower albedo and cause additional heat absorption and ice melt. [l.31]

2. **42: Cloud radiative processes dictated by cloud height, cloud optical thickness, and hydrometeor phase are also important in driving the radiative budget.**

   *Thank you for this remark. In the revised manuscript, we expanded the respective sentence to include a reference to the influence of cloud characteristics—such as height, optical thickness, and hydrometeor phase—on the radiative budget.*

   Cloud radiative processes, modulated by cloud height, optical thickness, and hydrometeor phase, are additional key drivers of the local energy balance and interact with LSP-induced atmospheric variability (Wang et al., 2018). [l.47]

3. **52: The NAO is a redistribution of atmospheric mass between the subpolar and subtropical regions of the North Atlantic that one can capture the NAO using surface pressure data, as mentioned here, but can also be identified using geopotential height data.**

   *We have revised the sentence accordingly to reflect this broader perspective and have clarified that NAO-related variability can be captured throughout the troposphere, not only at the surface.*

The NAO reflects a redistribution of atmospheric mass between the Azores High and the Icelandic Low, capturing shifts in the strength and position of these pressure systems. It is typically represented by surface pressure differences but can also be identified using geopotential height anomalies, as its influence extends throughout the troposphere. The NAO describes climate variability in the North Atlantic sector, influencing temperature and precipitation patterns across Europe, North America, and North Africa (Hanna et al., 2022; Hurrell et al., 2003; Silva et al., 2022). [l.58]

4. **64: Thermal advections will not necessarily follow lines of constant geopotential height, but will follow lines of constant thickness. To a first order, this is a reasonable interpretation, but it is important to be careful when interpreting the height pattern as showing advection.**

   *Yes, thermal advection is more accurately related to thickness gradients rather than geopotential height. In the revised manuscript, we have clarified this by expanding the explanation accordingly, noting the limitations of using height fields to infer advection and emphasizing that our interpretation is a first-order approximation commonly used in synoptic analyses.*

   While this approach is commonly used to interpret synoptic-scale flow, it is important to note that thermal advection—i.e., the transport of warm or cold air—more precisely follows thickness gradients rather than height lines. Thus, patterns in geopotential height can indicate both flow direction and relative atmospheric temperature [l.73]

5. **65: It appears there is a missing word or phrase after "west in".**

   *Yes, "summer" was missing and added in the revised manuscript.*

6. **65-66: As large accumulation and ablation events are often driven by cyclonic events, it would be worth saying more about how low-frequency circulation drives the storm track and the subsequent impact on mass balance.**

   *We fully agree that persistent large-scale circulation patterns play an important role in shaping storm tracks and thereby influencing both accumulation and ablation processes on the Greenland Ice Sheet. We will address this point in the revised manuscript in the discussion section, where we reflect on broader implications and future directions of our work. We consider this an appropriate place to include the connection between LSP persistence, storm activity, and surface mass balance, as we intend to expand our future analyses to explicitly include melt processes on the GrIS.*

   The persistence of large-scale atmospheric circulation patterns plays a central role in shaping storm tracks that influence accumulation and ablation over the GrIS. Persistent cyclonic or anticyclonic configurations can deflect or intensify the North Atlantic storm track, modulating both the frequency and intensity of moisture transport. These patterns impact surface mass balance not only by controlling snowfall during accumulation events, but also through warm-air advection, changes in cloud cover, and altered radiative forcing. [l.414]

7. **114: Why not use a standard 30-year reference period, such as 1981-2010?**

*Thank you for the suggestion. To incorporate recent warming as much as possible, we chose the most recent 30-year period available in our dataset (1986–2015). Since the 1991–2020 period—commonly used in current climatological assessments—is warmer than 1981–2010, our aim was to include as much of this warming signal as feasible within the constraints of homogeneous reanalysis data.*

8. **153: Is there any issue including a domain that reaches the pole?**

*Thank you for this great question. In our analysis, we did not encounter any issues related to including a domain extending near the pole. We accounted for the decreasing size of grid cells at higher latitudes by applying latitude-based weighting using the cosine of the latitude. That technically leads to the exclusion of the pole (cosinus(90°) = 0) and the domain only reaches up to 89.5°N (i.e., 90° minus half the 2° resolution of the 20CRv3 dataset). This approach ensures the reliability of the results without introducing artefacts from overestimating the impact of the region near the pole.*

9. **180: Are the warmest 15% "abnormal"?**

*We agree that the term "abnormal" may have been misleading in this context. To clarify, we refer to the warmest and coldest 15% of days based on the rank of their AT anomalies, without reference to a fixed climatological threshold. We have therefore adjusted the wording in the manuscript to avoid confusion and now refer simply to "the warmest/coldest 15% of all days."*

10. **220-224: It's not just the direction of flow that's important, but the geopotential height is a function of the thickness of the lower troposphere, which is a function of the mean (virtual) temperature of that layer. If, like pattern 5, you are sitting under a trough, you would expect colder than normal weather. The patterns may also represent potential storm tracks, such as LSP6, which I expect would be more conducive to storms traveling up Baffin Bay and affecting the west coast of Greenland.**

*Thank you for the insightful comment and for helping expand the interpretation of the identified patterns. We fully agree with your considerations. In the revised manuscript, we propose to incorporate this perspective in two places: first, by adding a sentence to the description of how geopotential height fields relate to both flow direction and thermal structure (along the lines of comment 4); and second, by including a more detailed interpretation of selected LSPs in the discussion section, highlighting their potential to indicate storm tracks and associated thermal anomalies.*

Thus, patterns in geopotential height can indicate both flow direction and relative atmospheric temperature. [l.75]

The geopotential height field also reflects the mean virtual temperature of the tropospheric layer, where lower heights typically indicate colder air masses. For

instance, the cold anomaly associated with LSP5 aligns with its position under a pronounced trough. Moreover, certain LSPs, such as LSP6, may represent configurations that support enhanced storm activity along common cyclone tracks— e.g., up Baffin Bay—which could contribute to temperature anomalies and variability along Greenland's west coast. [l.388]

**11. 265: I really like this figure, especially the heat map and persistence plot! Very helpful to understand the patterns.**

*Thank you for your positive feedback.*

**12. 319-324: I had been wondering why 20CR3 was used instead of the ERA-20C product when the 20CR3 product was first mentioned. This should be discussed earlier in the manuscript.**

*The other reviewer raised similar concerns and in the revised manuscript, we moved l. 317-320 to the end of the suggested section. So that the decision to choose 20CRv3 based on the prior study is in the method part with minor adjustments to embed it in the text. See the following:*

The approach follows the method described in Abermann et al. (2023). They evaluated the performance of the 20CRv3 and CERA-20C reanalysis models for two non-assimilated stations within the study area. Both models were interpolated and adjusted to the station altitudes, and their findings indicate that 20CRv3 aligns more closely with observations from 1930 and 1931 than CERA-20C. For this reason, we selected 20CRv3 for our analysis. Slivinski et al. (2021) confirmed the general good agreement between 20CRv3 and measured temperature. [l.119]

*We left the comparison of the reanalysis at WEG_L and the observations at the weather stations in the discussion part, as this is discussed further in the following paragraph.*

**349-350: It might be helpful to produce composite AT and SST anomalies (like Fig 2b,c) for each of the 8 patterns (overall and during each of the two periods studies).**

*Thank you for this constructive suggestion. We agree that composite AT and SST anomalies for each of the eight patterns—both overall and split by the two warming periods—would provide additional insight into the spatial climate signals associated with the LSPs. However, implementing this analysis would substantially expand the scope of the current manuscript, requiring a detailed and systematic evaluation beyond our current focus on linking LSPs to local AT anomalies and warming periods. Given the additional complexity, we consider this a valuable direction for future work and will keep it in mind for follow-up studies.*

**Answer review comment #3**

**Major comments**

**1. The number of SOM nodes (cluster centres) is only eight. The authors give an argument that they preferred to avoid the need the carry out two SOM analyses, first using a larger number of nodes and then regrouping to smaller number. However, the regrouping is not necessarily needed, and the number of nodes should be large enough to allow robust conclusions. In the results obtained, some SOM nodes (particularly 1, 3 and 8) may result in either warm or cold conditions at the study site. This seems to be attributed to the role of local processes. I agree that local processes often have a strong impact on near-surface air temperatures, particularly in regions of complex orography. However, the non-systematic sign of the temperature anomaly for a single SOM node may also result from the fact the smaller is the number of nodes, the larger is the variability of circulation patters within the a single node. This makes it impossible to understand (based on the analyses made here) what is the relative importance of small-scale processes and the variations between large-scale patterns within a SOM node. I suggest performing a new SOM analysis with a larger number of nodes (e.g., 20 or even more).**

*Thank you for this constructive comment. We acknowledge the trade-off between the number of SOM nodes and the interpretability of the resulting patterns. We tested several SOM configurations with both fewer and more clusters (6, 7, 9, 10, 15, 20, 25, and 30 nodes) also mentioned by a similar comment from Reviewer #1. While increasing the number of nodes naturally altered the distribution of patterns, it did not result in a clearer or more meaningful differentiation in terms of associated AT anomalies—particularly for zonal flow regimes. For example, a tangible split of the dominant zonal pattern (LSP 3 of the original SOM analysis in the manuscript) only emerged at 30 nodes, and even then, both clusters showed very similar temperature anomaly signatures.*

*In line with the purpose of our analysis—capturing the most dominant and interpretable large-scale circulation patterns affecting Greenland's temperature—the 8-node configuration provided a robust balance between generalization and detail. We emphasize that the SOM settings are somewhat arbitrary and depend on study goals and that increasing the number of nodes does not necessarily yield better insight into surface impacts due to internal pattern variability.*

*To further address the concern and for transparency, we added additional figures (see Fig. R5 and R6, now included in appendix A1 as Fig. A1.1 and A1.2) showing the SOM configuration with 20 nodes. The associated analysis on temperature anomalies yield similar conclusions to the main 8-node configuration – i.e., the majority of cases appears rather neutral, some stick out as positive anomalies, some as negative ones. One constraint is, that the increased number of nodes reduces the sample size per pattern considerably (e.g.,147/23/9 days with LSP 2 for the full study period 1900-2015/WP1/WP2 respectively), which in turn affects the statistical robustness of anomaly associations and hampers interpretability.*

*We hope this clarified our rationale for selecting 8 nodes based on interpretability and statistical significance across LSPs.*

*We included the example of 20 cluster centres in the appendix A1 of the revised manuscript, as well as the following sentence to address this point.*

We performed test runs for different number of clusters (see appendix A1), but  we aimed to avoid the need for subgrouping after applying the SOM algorithm, opting instead for a straightforward and consistent clustering approach. [l. 176]

In the following we show different parameters we tested for the SOM method. We tested different number of nodes – 6,7,8,9,10,15,20,25,30. As an example we added here the results with 20 nodes. As we wanted to avoid the manual regrouping after SOM as done by Schmidt et al. (2023) and Schuenemann and Cassano (2009) we concluded that eight clusters are 485 sufficient to show the expected large-scale patterns over Greenland and that they show the expected warm and cold patterns. More clusters do not result in more details. We further found that the key results are not sensitive to the number of clusters. [l.468]

[Figure]

Figure R5: Geopotential height of the 500 hPa pressure level of the detected LSPs using 20 nodes instead of 8 nodes as applying the SOM algorithm. The study site is marked with a green dot. For visual clarity, the SOM patterns are displayed in a 2D matrix;

however, the underlying topology is one-dimensional, with neighbourhood relations applying only sequentially along a single line from the top left to the bottom right, i.e., following the numbering of the LSPs. (This is Fig. A1.1. in the revised manuscript)

[Figure]

Figure R6.: Summary of the evaluation of the LSPs examined across the study periods applying 20 nodes instead of 8 nodes. The study periods are color-coded throughout the plot as follows: WP1 (1922–1932) in orange, WP2 (1993–2007) in green, and the full period (1900–2015) in blue. (a) Distribution of relative occurrence of each LSP across the study periods. (b) Average AT anomaly per LSP, with markers indicating the mean anomaly and whiskers representing ±1 standard deviation. (c) Distribution of persistence in days per LSP, with bold lines indicating mean lengths. The full period is at the top and both WPs at the bottom. (d) Annual average AT anomaly per LSP, with coloured frames representing the study periods. (This is now Fig. A.1.2 in the revised manuscript.)

**2. The SOMs are grouped in a one-dimensional array. Although the plots in Figure 3 are not shown in a row, it is evident that the analysis is made for an 1D array. This is**

**demonstrated by the fact that neighboring nodes (e.g., 1 and 4, 2 and 5, 3 and 6, 5 and 8) differ a lot from each other, which is not the case for a 2D SOM array. Using a 1D array is very unusual in the field of meteorology. In general, 2D SOM is preferred for most applications, in particular when it is relevant to visualize clusters or relationships, and the data does not have a natural linear ordering. A 1D SOM may be relevant for simple, linear datasets or specific ranking tasks, but it is less interpretable for general clustering or visualization. I suggest that the new SOM analysis will be made using a 2D array (e.g., 5 x 4 nodes).**

*Thank you for pointing this out. We agree that 2D SOMs are often used in meteorological applications, particularly for visualizing relationships between clusters in an intuitive spatial manner. However, the choice between a 1D and 2D grid topology depends on the study's goals, the nature of the dataset, and the design philosophy behind the SOM implementation.*

*In our study, we applied the SOM approach as proposed by Doan et al. (2021), which utilizes a 1D array of SOM nodes for meteorological data. While a 1D SOM may appear less conventional in meteorological clustering studies, it is not theoretically inferior. In the current revision process, we also got in touch with Assoc. Prof Doan (personal communication, 2025) in order to address this issue After careful consideration we point out: the original intent of our SOM application was dimensionality reduction and visualization, not necessarily the derivation of associations among different SOMs. In this case, our goal was not to interpret neighbouring nodes as gradual transitions but to identify and interpret distinct LSPs, each of which represent a typical configuration during individual periods. The SOM topology serves here primarily as a classification tool rather than a continuum and our intention is to assess relative distributions and anomalies of individual classes to compare them but not interpret the transition between LSPs.*

*Furthermore, other studies have also successfully employed 1D SOMs in atmospheric science applications using the same or similar methodology (e.g., Luong et al. (2024); Xie et al. (2025)), indicating that this approach has precedence and is suitable in contexts where the topological relationships among clusters are not the primary analytical focus.*

*We agree that a 2D SOM structure would be more suitable for exploring variability, transitions, and persistence among patterns, and we will consider this approach in future work to investigate inter-pattern relationships more comprehensively. However, in the context of our present work, we emphasize distinct circulation types during warming periods and therefore chose a 1D topology for its simplicity and interpretability.*

*To address the reviewer's concern and ensure transparency, we now clarify the 1D nature of the SOM.*

We use the 1D SOM method by Doan et al., (2021), applying their  distance function (structural similarity) and learning rate (0.1) as proposed by them.  [l.162]

*We also explicitly state that the 2D arrangement of patterns in Fig. 3 is purely for visual clarity with the following statement to the caption of the figure.*

For visual clarity, the SOM patterns are displayed in a 2D matrix; however, the underlying topology is one-dimensional, with neighbourhood relations applying only sequentially along a single line from the top left to the bottom right, i.e., following the numbering of the LSPs.

**3. The SOM analysis is made for a rather small area around Greenland (Figure 3). Even if certain circulation patters appear rather similar in the study area, fitting into a same SOM node, they may include considerable differences outside the boundaries of the study area. This may result in large differences in the transports of heat and moisture to Greenland. I cannot be sure if the effects on the results and conclusions of this study are large, but it is possible. Hence, I recommend that in the new SOM analyses the study area is enlarged.**

*Thank you for raising this important point. We agree that circulation patterns outside the immediate vicinity of Greenland affect heat and moisture transport—particularly in situations involving far-range advection (e.g., atmospheric rivers). However, some choice needs to be taken, hence, we applied length scales relevant for LSPs (i.e., several 100s of km).*

*To address your concern, we added a supplementary SOM analysis using a larger domain and include the resulting figures to the appendix of the revised manuscript (Fig. A1.3-6). These confirm that while including a broader area smooths the identified LSPs somewhat, it does not yield fundamentally different circulation types or significantly alter the interpretation over Greenland. Therefore, we retain the original domain in the main analysis for consistency and focus but acknowledge that future studies on specific phenomena (e.g., atmospheric rivers) may benefit from an extended spatial domain.*

We tested different domain sizes for the SOM analysis, including domains extending further south to 30°N and even covering the entire Northern Hemisphere. However, these broader domains produced less meaningful patterns in the context of Greenland, often introducing LSPs with little relevance to regional conditions. The domain selected for our main analysis—spanning 0–90°W and 55–90°N—captures the synoptic-scale circulation most relevant for Greenland. To illustrate the effect of domain choice, we also show results from a slightly extended domain (120°W to 20°E and 50°N to 90°N) using both eight and 20 SOM nodes. While using more clusters may offer finer distinctions, it also introduces interpretation challenges: for example, LSP 19 and 20 would need to be regrouped to allow for meaningful conclusions. Additionally, some patterns occur very rarely—LSP 19 appears on only 138/12/26 days (0.3/0.3/0.5%) during the full study period (1900–2015), WP1, and WP2 respectively—limiting the statistical robustness of their associated AT anomalies. [l.473]

*We also included the following in the discussion to include the important point raised by you.*

While our selected domain effectively captures the dominant LSPs over Greenland, future studies could benefit from an expanded and adjusted spatial domain to further resolve specific influences - such as atmospheric rivers - that may significantly affect surface climate and mass balance. [l.421]

[Figure]

Figure R7: Geopotential height of the 500 hPa pressure level of the eight LSPs as defined by SOM with a larger input domain of 120°W -20°E and 50-90°N. The study site is marked with a green dot. For visual clarity, the SOM patterns are displayed in a 2D matrix; however, the underlying topology is one-dimensional, with neighbourhood relations applying only sequentially along a single line from the top left to the bottom right, i.e., following the numbering of the LSPs. (This is Fig. A.1.3 in the revised manuscript)

[Figure]

Figure R8: Summary of the evaluation of the eight LSPs examined across the study periods for a larger domain of 120°W -20°E and 50-90°N. The study periods are color-coded throughout the plot as follows: WP1 (1922–1932) in orange, WP2 (1993–2007) in green, and the full period (1900–2015) in blue. (a) Distribution of relative occurrence of each LSP across the study periods. (b) Average AT anomaly per LSP, with markers indicating the mean anomaly and whiskers representing ±1 standard deviation. (c) Distribution of persistence in days per LSP, with bold lines indicating mean lengths. The full period is at the top and both WPs at the bottom. (d) Annual average AT anomaly per LSP, with coloured frames representing the study periods. (This is Fig. A1.4 in the revised manuscript.)

[Figure]

Figure R9: Geopotential height of the 500 hPa pressure level of the 20 LSPs as defined by SOM with a larger input domain of 120°W -20°E and 50-90°N. The study site is marked with a green dot. For visual clarity, the SOM patterns are displayed in a 2D matrix; however, the underlying topology is one-dimensional, with neighbourhood relations applying only sequentially along a single line from the top left to the bottom right, i.e., following the numbering of the LSPs.  (This is Fig. A1.5 in the revised manuscript.)

[Figure]

Figure R10:: Summary of the evaluation of the 20 LSPs examined across the study periods for a larger domain and 20 cluster centres. The study periods are color-coded throughout the plot as follows: WP1 (1922–1932) in orange, WP2 (1993–2007) in green, and the full period (1900–2015) in blue. (a) Distribution of relative occurrence of each LSP across the study periods. (b) Average AT anomaly per LSP, with markers indicating the mean anomaly and whiskers representing ±1 standard deviation. (c) Distribution of persistence in days per LSP, with bold lines indicating mean lengths. The full period is at the top and both WPs at the bottom. (d) Annual average AT anomaly per LSP, with coloured frames representing the study periods. (This is Fig. A1.6 in the revised manuscript)

**4. The analysis addresses near-surface air temperatures with focus on coastal site at an elevation of 940 m. There the typical atmospheric pressure may be roughly 910–920 hPa. However, the analysis is made on the basis of 500 hPa geopotential height fields. These indeed well characterize the large-scale circulation patterns, but in a baroclinic atmosphere the wind vector at the 500 hPa may deviate a lot from that at the 910.920 hPa level. This should be discussed when making conclusions on the role heat advection associated with various SOM nodes. Over a flat surface, the effect of large-scale circulation**

**on 2-m air temperature might be best analysed on the basis of 850-hPa fields but in Greenland this is naturally liable to errors. However, the authors could consider, if the SOM analysis would be better to do for 700 instead of 500 hPa fields.**

*We thank the reviewer for this valuable comment and fully agree that the 500 hPa geopotential height fields primarily characterize the large-scale circulation, while the near-surface wind vectors may differ significantly due to baroclinicity and topographic effects. As the reviewer notes, 850 hPa or 700 hPa fields could potentially offer different approximations of thermal advection closer to the surface. However, our intention was explicitly to capture the synoptic-scale structure of circulation without the confounding effects of topography, which is particularly complex in Greenland. To put this in context, the immediate topography around our station at 940 m a.s.l. reaches ice-free areas up to more than 1000 m a.s.l., and the ice sheet rises considerably higher. Therefore, choosing higher pressure fields would involve intersecting those with the topography or getting close to near-surface impact that may again create interpretation features that do not relate to the large-scale synoptic conditions but rather local factors.*

*Therefore, we chose the 500 hPa level because it is detached from surface influences and represents the large-scale flow regime that governs the general direction and origin of air masses.*

**5. I recommend carrying out seasonal analyses, as a certain geopotential height pattern may have very different effects on near-surface air temperatures in winter and summer.**

*We appreciate the reviewer's suggestion to carry out seasonal analyses, and we agree that the effect of individual LSPs can differ significantly between seasons due to variations in solar radiation, surface conditions, and atmospheric stratification. While the primary goal of our study is to characterize the overall relationship between circulation patterns and AT anomalies, we have now included a seasonal breakdown of AT anomalies per LSP to illustrate this variability more explicitly, also very much in line with comments by the others reviewers This additional analysis serves as a complement to our annual perspective and is presented in the revised manuscript as an extension of Fig. 4b. It confirms that while the overall patterns are robust, their local thermal impacts vary in intensity and sometimes sign between seasons.*

*We appreciate the reviewer's suggestion to carry out seasonal analyses, we agree that seasonal variations in circulation can affect near-surface air temperatures differently, and this is an important consideration. Our study focuses on identifying dominant LSPs associated with regional warming over Greenland during two warming periods (WP1 and WP2), irrespective of season, as a first step toward understanding long-term changes in AT–LSP relationships.*
*To address the seasonal component without recomputing the SOMs seasonally, we performed complementary analyses. Specifically:*

1. *Seasonal differences in near-surface temperature trends are shown in Fig. A2.1. These confirm that warming is strongest in winter during both WPs, in agreement with previous findings (e.g., Box et al., 2009).*

2. *Seasonal variations in LSP occurrence are examined in Fig. A2.2. This demonstrates that the main patterns we identify occur across all seasons, although with different frequencies.*
3. *While we acknowledge that SOMs computed for each season separately might reveal more distinct circulation structures, they would address a different research question—namely, identifying season-specific LSPs and their regional temperature effects. Our current aim is to characterize the most dominant large-scale features relevant to warming events over the entire year.*

*Nonetheless, we see seasonal SOMs as a valuable direction for future work, particularly to better understand transitional behaviour or seasonally dominant patterns.*

**6. I am not convinced if it is an optimal approach to focus on a single study site (WEG_L).**

*Following comments by other reviewers we acknowledge that the motivation for our study has not been entirely clear to the reader, which we expand upon in the revised version. The point-scale of WEG_L serves as the motivation to study the large-scale synoptic conditions and solely determines the centre of the spatial domain for which we relate the LSPs to AT anomalies at the surface.*

*Hence, the selection of WEG_L was intentional, as it represents a well-documented, and relatively high-elevation site in the ablation zone, close to the ice margin. Our objective was to conduct a detailed case study to test whether and how large-scale circulation patterns are reflected in local AT anomalies and to establish a framework that can be extended to additional locations in future work.*

*Given the spatial scope of the SOM-derived LSPs, the broader circulation–temperature relationships established here are likely applicable to other regions of Greenland. Clearly, we see the inclusion of multiple sites as a natural and important step for follow-up research.*

*To address the reviewers' concern that this connection is unclear, we added following explanation to the revised manuscript.*

Using the same location as Wegener's expedition allows for a direct comparison between past and present atmospheric conditions, enabling a better understanding of long-term changes and the role of large-scale atmospheric patterns (LSPs) in shaping regional climate variability.  [l.43]

From a climatological perspective, WEG_L is located in a region where AT changes are among the strongest during both WPs, as seen in Fig. 2 (b) and (c). The particular strong and significant warming in this area further supports WEG_L as a representative study site, making it a well-suited choice for this study. [l.230]

**Minor comments**

**Lines 65-66: "cyclones following the North Atlantic Oscillation" is unclear expression.**

*Indeed, this was a mistake from our side. It was supposed to be the North Atlantic Current, but we adjusted the sentence as follows to avoid unclear expressions.*

Other synoptic patterns involve the preferred track of cyclones  northward through the Davis Strait and Baffin Bay  or move northwards following the west coast. [l.77]

**Line 108: 6.5 K/1000 m**

*Thank you for spotting this mistake and giving us the chance to correct it.*

**Line 217: Add southwest?**

*Yes, thank you.*

**Line 381: Should the phase of Arctic Oscillation be somehow seen in changes in the occurrence of various SOM nodes?**

*The Arctic Oscillation impacts air circulation and advection and might affect LSP distribution. This statement is just given as an outlook of the analysis as this is connected to the reason for the occurrence of LSPs – a question worth investigating in the future to assess how the change of certain drivers can influence LSP distribution. Based on a comment of reviewer#1 we added the following sentence as additional explanation:*

The variability of LSP occurrences may also reflect broader atmospheric circulation changes, potentially influenced by climate modes such as the Arctic Oscillation (AO). A strong polar vortex, often associated with a positive AO phase, can contribute to more stable, zonal airflow patterns, while a weaker vortex during a negative AO phase allows for increased meandering and variability in geopotential height fields. [l.376]

**Lines 398-400: Also the increase in CO2 concentration matters.**

*Excellent point, we added it in the list of influences:*

Instead, our findings imply that the local AT is influenced by both the specific LSP characteristics and potentially other climate mechanisms, such as Arctic feedback loops, other factors detached from atmospheric drivers (e.g., sea ice occurrence), increase in CO2 concentration and changes in global circulation. [l.449]

**Line 403: globally?**

*The sentence refers to the fact that WP2 (1979–2020) occurred during a period when temperatures were rising globally. In contrast, WP1 (1915–1945) is part of the early twentieth-century warming, which was more pronounced in the Arctic than in other regions and less globally uniform. Therefore, the elevated ATs in WP2 likely reflect the stronger and more widespread background warming, in addition to specific circulation patterns. We therefore suggest to keep 'globally' but added the following to avoid confusion for the readers:*

This suggests that background warming may enhance an increase of AT regardless of LSP type, especially in WP2, which occurred during a period of more globally uniform warming, in contrast to WP1, where warming was more regionally concentrated in the Arctic. [l.451]

**Literature:**

Abermann, J., Vandecrux, B., Scher, S., Löffler, K., Schalamon, F., Trügler, A., Fausto, R., & Schöner, W. (2023). Learning from Alfred Wegener's pioneering field observations in West Greenland after a century of climate change. *Scientific Reports*, *13*(1), 7583. https://doi.org/10.1038/s41598-023-33225-9

Box, J. E., Yang, L., Bromwich, D. H., & Bai, L. S. (2009). Greenland ice sheet surface air temperature variability: 1840-2007. *Journal of Climate*, *22*(14), 4029–4049. https://doi.org/10.1175/2009JCLI2816.1

Cappelen, J., Vinther, B. M., Kern-Hansen, C., Laursen, E. V., & Jørgensen, P. V. (2021). *Greenland-DMI Historical Climate Data Collection 1784-2020*. *May*, 105. www.dmi.dkwww.dmi.dkurlhttps://www.dmi.dk/publikationer/www.dmi.dk

Doan, Q. Van, Kusaka, H., Sato, T., & Chen, F. (2021). S-SOM v1.0: A structural self-organizing map algorithm for weather typing. *Geoscientific Model Development*, *14*(4), 2097–2111. https://doi.org/10.5194/gmd-14-2097-2021

Fettweis, X., Hanna, E., Lang, C., Belleflamme, A., Erpicum, M., & Gallée, H. (2013). Brief communication Important role of the mid-tropospheric atmospheric circulation in the recent surface melt increase over the Greenland ice sheet. *Cryosphere*, *7*(1), 241–248. https://doi.org/10.5194/TC-7-241-2013

Hanssen-Bauer, I., Førland, E., Hisdal, H., Mayer, S., Sandø, A. B., Sorteberg, A., Adakudlu, M., Andresen, J., Bakke, J., Beldring, S., Benestad, R., Bilt, W., Bogen, J., Borstad, C., Breili, K., Breivik, Ø., Børsheim, K. Y., Christiansen, H. H., Dobler, A., … Wong, W. K. (2019). Climate in Svalbard 2100 – a knowledge base for climate adaptation. In *Norsk klimaservicesenter (NKSS)/Norwegian Centre for Climate Services (NCCS)* (Issue 1). https://doi.org/http://dx.doi.org/10.25607/OBP-888

Hartl, L., Schmitt, C., Wong, T., Vas, D. A., Enterkin, L., & Stuefer, M. (2023). Long-Term Trends in Ice Fog Occurrence in the Fairbanks, Alaska, Region Based on Airport Observations. *Journal of Applied Meteorology and Climatology*, *62*(9), 1263–1278. https://doi.org/10.1175/JAMC-D-22-0190.1

Hartl, L., Stuefer, M., & Saito, T. (2020). The mountain weather and climate of Denali, Alaska—an overview. *Journal of Applied Meteorology and Climatology*, *59*(4), 621–636. https://doi.org/10.1175/JAMC-D-19-0105.1

Hermann, M., Papritz, L., & Wernli, H. (2020). A Lagrangian analysis of the dynamical and thermodynamic drivers of large-scale Greenland melt events during 1979-2017. *Weather and Climate Dynamics*, *1*(2), 497–518. https://doi.org/10.5194/wcd-1-497-2020

Hofsteenge, M. G., Cullen, N. J., Sodemann, H., & Katurji, M. (2024). Synoptic drivers and moisture sources of snowfall in coastal Victoria Land, Antarctica. *ESS Open Archive* , *Vl*, 1–20. 10.22541/essoar.172286685.53525508/v1

Luong, T. M., Dasari, H. P., Doan, Q. Van, Alduwais, A. K., & Hoteit, I. (2024). Organized precipitation and associated large-scale circulation patterns over the Kingdom of Saudi Arabia. *International Journal of Climatology*, *44*(10), 3295–3314. https://doi.org/10.1002/joc.8524

Mattingly, K. S., Mote, T. L., & Fettweis, X. (2018). Atmospheric River Impacts on Greenland Ice Sheet Surface Mass Balance. *Journal of Geophysical Research: Atmospheres*, *123*(16), 8538–8560. https://doi.org/10.1029/2018JD028714

Mioduszewski, J. R., Rennermalm, A. K., Hammann, A., Tedesco, M., Noble, E. U., Stroeve, J. C., & Mote, T. L. (2016). Atmospheric drivers of Greenland surface melt revealed by self-organizing maps. *Journal of Geophysical Research: Atmospheres*, *121*(10), 5095–5114. https://doi.org/10.1002/2015JD024550

Neff, W., Compo, G. P., Ralph, F. M., & Shupe, M. D. (2014). Continental heat anomalies and the extrememelting of the Greenland ice surface in 2012 and 1889. *Journal of Geophysical Research*, *175*(4449), 238. https://doi.org/10.1038/175238c0

Preece, J. R., Wachowicz, L. J., Mote, T. L., Tedesco, M., & Fettweis, X. (2022). Summer Greenland Blocking Diversity and Its Impact on the Surface Mass Balance of the Greenland Ice Sheet. *Journal of Geophysical Research: Atmospheres*, *127*(4), e2021JD035489. https://doi.org/10.1029/2021JD035489

Schmid, T., Radi, V., Tedstone, A., Lea, J. M., Brough, S., Hermann, M., Radić, V., Tedstone, A., Lea, J. M., Brough, S., & Hermann, M. (2023). Atmospheric drivers of melt-related ice speed-up events on the Russell Glacier in Southwest Greenland. *Cryosphere*, *17*(January), 1–32. https://doi.org/10.5194/tc-17-3933-2023

Schmidt, L. S., Schuler, T. V, Thomas, E. E., & Westermann, S. (2023). Meltwater runoff and glacier mass balance in the high Arctic: 1991-2022 simulations for Svalbard. *EGUsphere*, *2023*, 1–32. https://doi.org/10.5194/egusphere-2022-1409

Schuenemann, K. C., & Cassano, J. J. (2009). Changes in synoptic weather patterns and Greenland precipitation in the 20th and 21st centuries: 1. Evaluation of late 20th century simulations from IPCC models. *J. Geophys. Res. Atmos.*, *114*(D20), 20113. https://doi.org/10.1029/2009JD011705

Wang, W., Zender, C. S., & van As, D. (2018). Temporal Characteristics of Cloud Radiative Effects on the Greenland Ice Sheet: Discoveries From Multiyear Automatic Weather Station Measurements. *Journal of Geophysical Research: Atmospheres*, *123*(20), 11,348-11,361. https://doi.org/10.1029/2018JD028540

Wickström, S., Jonassen, M. O., Cassano, J. J., & Vihma, T. (2020). Present Temperature, Precipitation, and Rain-on-Snow Climate in Svalbard. *Journal of Geophysical Research: Atmospheres*, *125*(14), e2019JD032155. https://doi.org/10.1029/2019JD032155

Xie, X., Qin, D., Xiao, M., & Lin, K. (2025). Urbanization Enhances Shorter-Duration Precipitation Intensity in the Yangtze River Delta Region. *Journal of Geophysical Research: Atmospheres*, *130*(8), 1–12. https://doi.org/10.1029/2024JD043300

---

## Author Response (AR2)

Dear editor,

Thank you for your message and for the time and effort you and the reviewers have dedicated to handling our manuscript. We greatly appreciate the thoughtful and constructive feedback throughout the review process.

We have addressed the requested minor revisions as follows:

1. The abstract has been revised to more specifically highlight the key findings of our analysis, with a clearer description of the differences in large-scale atmospheric patterns during the two recent Greenland warming periods.
2. The figures and text previously included in the Appendix have been moved to a Supplementary file, in line with your recommendation.
3. Additionally, we have corrected the terminology throughout the manuscript from "atmospheric large-scale pattern" to "large-scale atmospheric pattern." These changes are reflected in the version with tracked modifications.

All revised documents have been uploaded accordingly.

On behalf of all co-authors, I would like to express our gratitude to you and the reviewers for your valuable contributions to improving the manuscript.

Kind regards,

Florina Schalamon